# *OmniWorld*: A Multi-Domain and Multi-Modal Dataset for 4D World Modeling

**Yang Zhou**[1,2] **Yifan Wang**[2,3] **Jianjun Zhou**[2,4,5] **Wenzheng Chang**[2,3] **Haoyu Guo**[2] **Zizun Li**[2,6]
**Kaijing Ma**[1,2] **Xinyue Li**[2,3] **Yating Wang**[2,7] **Haoyi Zhu**[2,6] **Mingyu Liu**[2,4] **Dingning Liu**[1,2] **Jiange Yang**[2,8]
**Zhoujie Fu**[2,9] **Junyi Chen**[2,3] **Chunhua Shen**[4] **Jiangmiao Pang**[2] **Kaipeng Zhang**[2] **Tong He**[2,5,†]

[1]Fudan University  [2]Shanghai AI Laboratory  [3]Shanghai Jiao Tong University  [4]Zhejiang University
[5]Shanghai Innovation Institute  [6]University of Science and Technology of China  [7]Tongji University
[8]Nanjing University  [9]Nanyang Technological University  [†]Corresponding Author

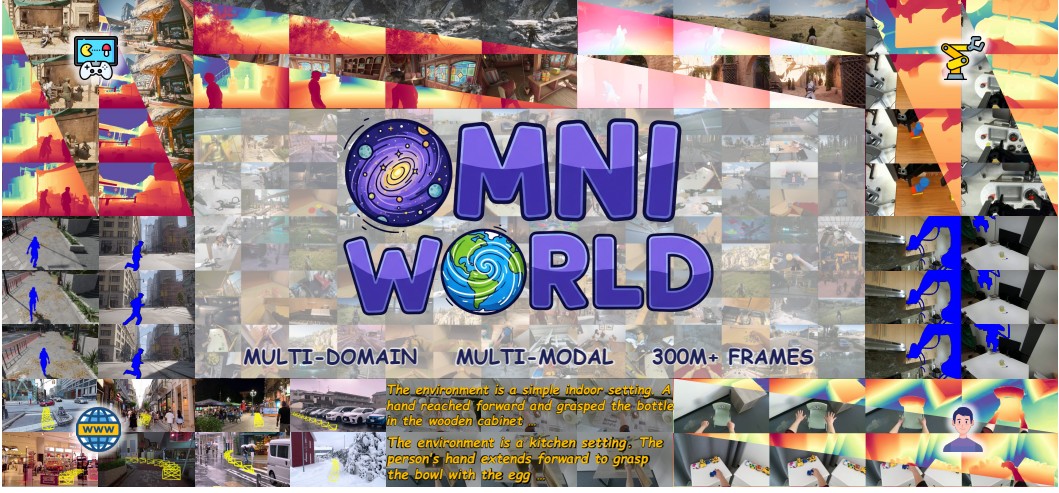

Figure 1: We introduce *OmniWorld*, a large-scale, multi-domain, and multi-modal dataset. *OmniWorld* provides a rich resource for 4D world modeling by integrating high-quality data from multiple domains and offers a variety of data types, including depth maps, camera poses, text captions, optical flow and foreground masks. *OmniWorld* is designed to accelerate the development of more general models for modeling the real physical world.

## ABSTRACT

The field of 4D world modeling—aiming to jointly capture spatial geometry and temporal dynamics—has witnessed remarkable progress in recent years, driven by advances in large-scale generative models and multimodal learning. However, the development of truly general 4D world models remains fundamentally constrained by the availability of high-quality data. Existing datasets and benchmarks often lack the dynamic complexity, multi-domain diversity, and spatial-temporal annotations required to support key tasks such as 4D geometric reconstruction, future prediction, and camera-controlled video generation. To address this gap, we introduce *OmniWorld*, a large-scale, multi-domain, multi-modal dataset specifically designed for 4D world modeling. *OmniWorld* consists of a newly collected *OmniWorld-Game* dataset and several curated public datasets spanning diverse domains. Compared with existing synthetic datasets, *OmniWorld-Game* provides richer modality coverage, larger scale, and more realistic dynamic interactions. Based on this dataset, we establish a challenging benchmark that exposes the limitations of current state-of-the-art (SOTA) approaches in modeling complex 4D environments. Moreover, fine-tuning existing SOTA methods on *OmniWorld* leads to significant performance gains across 4D reconstruction and video generation tasks, strongly validating *OmniWorld* as a powerful resource for training and evaluation. We envision *OmniWorld* as a catalyst for accelerating the development of general-purpose 4D world models, ultimately advancing machines' holistic understanding of the physical world. Project Page: https://yangzhou24.github.io/OmniWorld/

Table 1: **Comparisons between *OmniWorld-Game* and existing synthetic datasets.** *OmniWorld-Game* surpasses existing public synthetic datasets in modal diversity and data scale.

| Dataset | Scene Type | Motion | Resolution | # Frames | Depth | Camera | Text | Optical flow | Fg. masks |
|---|---|---|---|---|---|---|---|---|---|
| MPI Sintel (Butler et al., 2012) | Mixed | Dynamic | $1024 \times 436$ | 1K | ✔ | ✔ | ✗ | ✔ | ✔ |
| FlyingThings++ (Mayer et al., 2016; Harley et al., 2022) | Outdoor | Dynamic | $960 \times 540$ | 28K | ✔ | ✗ | ✗ | ✔ | ✔ |
| TartanAir (Wang et al., 2020) | Mixed | Dynamic | $640 \times 480$ | 1,000K | ✔ | ✔ | ✗ | ✔ | ✔ |
| BlendedMVS (Yao et al., 2020) | Mixed | Static | $768 \times 576$ | 17K | ✔ | ✔ | ✗ | ✗ | ✗ |
| HyperSim (Roberts et al., 2021) | Indoor | Static | $1024 \times 768$ | 77K | ✔ | ✔ | ✗ | ✗ | ✔ |
| Dynamic Replica (Karaev et al., 2023) | Indoor | Dynamic | $1280 \times 720$ | 169K | ✔ | ✔ | ✗ | ✔ | ✔ |
| Spring (Mehl et al., 2023) | Mixed | Dynamic | $1920 \times 1080$ | 23K | ✔ | ✔ | ✗ | ✔ | ✗ |
| EDEN (Le et al., 2021) | Outdoor | Static | $640 \times 480$ | 300K | ✔ | ✔ | ✗ | ✔ | ✔ |
| PointOdyssey (Zheng et al., 2023) | Mixed | Dynamic | $960 \times 540$ | 216K | ✔ | ✔ | ✗ | ✗ | ✔ |
| SeKai-Game (Li et al., 2025) | Outdoor | Dynamic | $1920 \times 1080$ | 4,320K | ✗ | ✔ | ✔ | ✗ | ✗ |
| ***OmniWorld-Game* (Ours)** | Mixed | Dynamic | $1280 \times 720$ | **18,515K** | ✔ | ✔ | ✔ | ✔ | ✔ |

# 1 INTRODUCTION

The development of world models (DeepMind, 2025; Ha & Schmidhuber, 2018; Agarwal et al., 2025; LeCun, 2022; Hafner et al., 2023) has become a central pursuit in visual intelligence systems, aiming to build systems that can simulate and reason about the physical world. This capability goes beyond simple static perception, demanding models that can simulate dynamic environments, predict object motion, infer causality, and generate content that adheres to physical laws. Such spatio-temporal modeling is a cornerstone for effective world models, with its development critically dependent on large-scale, multi-domain, and multi-modal datasets (Feng et al., 2024; Team et al., 2025a; Chen et al., 2025; He et al., 2025b; Team et al., 2025b; Yu et al., 2025b;a).

Two fundamental tasks that reflect a model's world modeling capability have drawn widespread attention: 3D geometric foundation models (Wang et al., 2024c; Leroy et al., 2024; Zhang et al., 2024; Yang et al., 2025a; Tang et al., 2024; Wang et al., 2025b; Zhang et al., 2025; Wang et al., 2025a;d), and camera-controlled video generation models (Wang et al., 2024d; He et al., 2024; Zheng et al., 2024; Bahmani et al., 2024; Bai et al., 2025; YU et al., 2025). The former aims to extract comprehensive 3D geometric information from 2D image inputs, while the latter focuses on generating dynamic video content that follows precise spatio-temporal instructions. Both tasks heavily rely on large-scale, high-quality datasets with rich modalities, including RGB images, depth maps, and camera poses.

However, existing benchmarks and datasets for evaluating and training these models have significant limitations. In the domain of 3D geometric foundation models, existing benchmarks suffer from short sequence lengths, which constrain the evaluation of a model's long-term robustness. For example, Sintel (Butler et al., 2012), which is a widely used dataset, consists of videos with an average length of only 50 frames. Furthermore, the limited motion amplitude and single-action types within these datasets (e.g., Bonn's (Palazzolo et al., 2019) focuses on indoor human motion, Kitti's (Geiger et al., 2013) focuses on outdoor street scenes) fail to comprehensively evaluate model performance in complex, dynamic environments. Similarly, in the field of camera-controlled video generation, mainstream datasets like RealEstate10K (Zhou et al., 2018) primarily consist of static scenes with smooth camera trajectories. This lack of diverse object motion and complex camera operations results in a noticeable gap between the dataset's content and real-world scenarios, thereby hindering a comprehensive assessment of a model's true capabilities.

From the perspective of training data, there is a critical scarcity of high-quality, multi-domain, multi-modal datasets that include rich geometric annotations. For instance, in image or video generation, while there are numerous image-text (Schuhmann et al., 2022; Gadre et al., 2023) or video-text datasets (Chen et al., 2024; Nan et al., 2024; Ju et al., 2024), they often lack critical geometric modalities such as depth maps, camera poses, and optical flow. Similarly, the demand for large-scale, diverse datasets with accurate geometric annotations is increasingly urgent for 3D geometric foundation models.

To address these shortcomings, we introduce *OmniWorld*, a large-scale, multi-domain, and multi-modal dataset composed of a self-collected high-quality *OmniWorld-Game* synthetic dataset and several public datasets. Its core characteristics are: **1) High-Quality 4D Data.** *OmniWorld-Game* is a massive synthetic video dataset comprising over 96K clips and more than 18M frames, with a total duration of over 214 hours. It is captured from diverse game environments with 720P RGB images, dense ground truth depth maps, accurate camera poses, and annotations for text captions, optical flow and foreground masks. As shown in Tab. 1, the dataset significantly surpasses existing

Table 2: *OmniWorld* **structure**. A smiling face (😊) indicates the modality is newly (re-)annotated by us, a green check (✔) denotes ground-truth data that already exists in the original dataset, and a red cross (✗) marks missing modalities.

| Dataset | Domain | # Seq. | FPS | Resolution | # Frames | Depth | Camera | Text | Opt. flow | Fg. masks |
|---|---|---|---|---|---|---|---|---|---|---|
| *OmniWorld-Game* | Simulator | 96K | 24 | 1280×720 | 18,515K | 😊 | 😊 | 😊 | 😊 | 😊 |
| AgiBot (Bu et al., 2025) | Robot | 20K | 30 | 640×480 | 39,247K | 😊 | ✔ | ✔ | ✗ | 😊 |
| DROID (Khazatsky et al., 2024) | Robot | 35K | 60 | 1280×720 | 26,643K | 😊 | ✔ | 😊 | 😊 | 😊 |
| RH20T (Fang et al., 2024) | Robot | 109K | 10 | 640×360 | 53,453K | ✗ | ✔ | 😊 | 😊 | 😊 |
| RH20T-Human (Fang et al., 2024) | Human | 73K | 10 | 640×360 | 8,875K | ✗ | ✔ | 😊 | ✗ | ✗ |
| HOI4D (Liu et al., 2022) | Human | 2K | 15 | 1920×1080 | 891K | 😊 | 😊 | 😊 | 😊 | ✔ |
| Epic-Kitchens (Damen et al., 2018) | Human | 15K | 30 | 1280×720 | 3,635K | ✗ | 😊 | 😊 | ✗ | ✗ |
| Ego-Exo4D (Grauman et al., 2024) | Human | 4K | 30 | 1024×1024 | 9,190K | ✗ | ✔ | 😊 | 😊 | ✗ |
| HoloAssist (Wang et al., 2023) | Human | 1K | 30 | 896×504 | 13,037K | ✗ | 😊 | 😊 | 😊 | ✗ |
| Assembly101 (Sener et al., 2022) | Human | 4K | 60 | 1920×1080 | 110,831K | ✗ | ✔ | 😊 | 😊 | 😊 |
| EgoDex (Hoque et al., 2025) | Human | 242K | 30 | 1920×1080 | 76,631K | ✗ | ✔ | 😊 | ✗ | ✗ |
| CityWalk (Li et al., 2025) | Internet | 7K | 30 | 1280×720 | 13,096K | ✗ | 😊 | ✔ | ✗ | ✗ |

public synthetic datasets in modal diversity and scale. **2) Multi-Domain Coverage.** By integrating datasets from four key domains including simulator, robot, human, and the internet, *OmniWorld* covers a wide range of real-world and virtual scenarios, greatly enhancing data diversity. **3) Multi-Modality Annotations.** *OmniWorld* provides a rich suite of multi-modal annotations, crucial for detailed world modeling, as shown in Tab. 2.

Based on *OmniWorld-Game*, we propose a new benchmark for both 3D geometric foundation models and camera-controlled video generation models. Our *OmniWorld-Game* benchmark provides challenging, complex scenarios and dynamics that accurately reflect a model's true world capabilities, revealing the limitations of current SOTAs. By fine-tuning existing SOTAs (e.g., DUSt3R (Wang et al., 2024c), CUT3R (Wang et al., 2025b), Reloc3r (Dong et al., 2024), AC3D (Bahmani et al., 2024)) with *OmniWorld*, we demonstrate significant performance improvements on public benchmarks. This strongly validates *OmniWorld* as a powerful training resource for enhancing world modeling capabilities.

In summary, our contributions are as follows:

1. We introduce *OmniWorld*, a multi-domain and multi-modal dataset designed to address the lack of diversity in existing datasets. Its self-collected subset, *OmniWorld-Game*, surpasses current synthetic datasets in both modality diversity and data volume.

2. We establish a comprehensive benchmark for 3D geometric foundation models and camera-controlled video generation models based on *OmniWorld-Game*, providing a unified platform for evaluation.

3. We fine-tune several SOTAs on *OmniWorld* and observe significant performance gains, underscoring its value as a training resource.

## 2 *OmniWorld* DATASET

### 2.1 DATA ACQUISITION

Our data acquisition strategy is centered on our novel, self-collected *OmniWorld-Game* dataset, which is strategically supplemented with curated data from three other distinct domains: robot, human, and internet, as illustrated in Fig. 2. This strategy allows us to integrate the strengths of diverse data sources to comprehensively capture real-world complexity.

**Simulator domain.** To acquire the high-precision and temporally consistent multi-modal data that is hard to obtain in the real world, we collect *OmniWorld-Game* from game environments. Following prior works (Richter et al., 2016; Yang et al., 2024a; Feng et al., 2024; Team et al., 2025a), we utilize ReShade (ReShade Contributors, 2024) to access depth information during the rendering process, and simultaneously capture synchronized RGB images from the screen using OBS (Contributors, 2024). This approach offers significant advantages: 1) High-Precision Modal Data. We can precisely control the environment and acquire accurate depth data, which is often unattainable

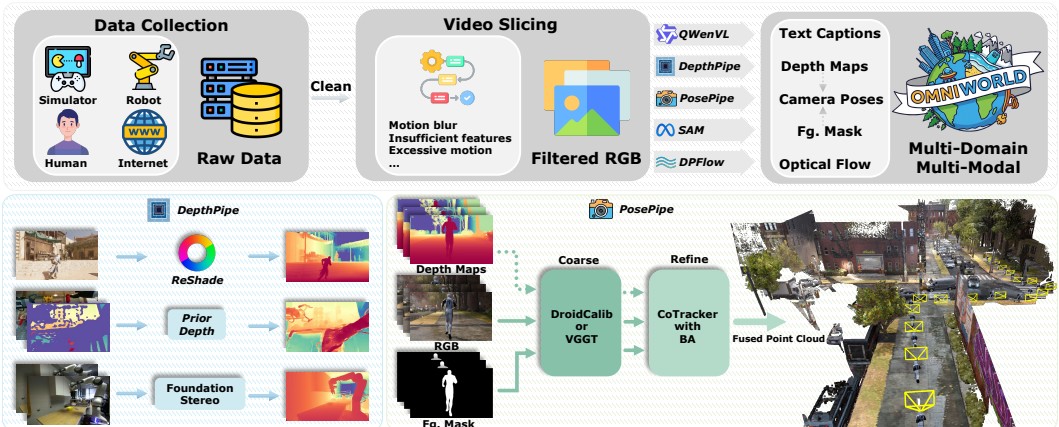

Figure 2: *OmniWorld* **acquisition and annotation pipeline.** We collect raw data from diverse domains and apply a video slicing filter to obtain high-quality RGB sequences. These sequences are then processed through a suite of specialized pipelines to generate multi-modal annotations, including text captions, depth maps, camera poses, foreground masks, and optical flow.

in real-world settings. 2) Rich Real-World Scene Simulation. Modern virtual environments provide highly realistic graphics and diverse simulations of real-world scenarios, such as complex settings from wilderness to urban areas.

**Robot domain.** We integrate public datasets from robot manipulation and human-robot interaction tasks, including AgiBot (Bu et al., 2025), DROID (Khazatsky et al., 2024), and RH20T (Fang et al., 2024). These datasets provide valuable sequences of robot-environment interactions and navigation, which are essential for tasks involving robotic manipulation and physical world understanding.

**Human domain.** We incorporate public datasets describing various human activities, including RH20T-Human (Fang et al., 2024), HOI4D (Liu et al., 2022), Epic-Kitchens (Damen et al., 2018), Ego-Exo4D (Grauman et al., 2024), HoloAssist (Wang et al., 2023), Assembly101 (Sener et al., 2022), and EgoDex (Hoque et al., 2025). These datasets capture diverse human behaviors, ranging from daily activities to complex assembly tasks, from both egocentric and exocentric perspectives.

**Internet domain.** To acquire large-scale, realistic, and diverse in-the-wild scene data, we utilize the CityWalk dataset (Li et al., 2025). It offers rich real-world street view videos from the internet.

To prepare the raw data, we perform video slicing to ensure high quality and temporal coherence. This crucial preprocessing step filters out unsuitable frames (e.g., those with motion blur or insufficient features) and segments long recordings into shorter, manageable clips. The resulting high-quality video segments are then passed to our multi-modal annotation pipeline.

## 2.2 DATA ANNOTATION

We primarily annotate the following key modalities: depth maps, camera poses, text captions, optical flow, and foreground masks (see Fig. 2 for the overall pipeline). Here we briefly introduce the annotation method of each modality.

**Depth maps.** Accurate depth information is paramount for geometric modeling. To ensure the quality and consistency of depth maps, we adopt a tailored approach based on the data source. For the self-collected dataset *OmniWorld-Game*, as mentioned in Sec. 2.1, we directly access depth information during the rendering process using tools like ReShade (ReShade Contributors, 2024).

For public datasets like AgiBot (Bu et al., 2025) and HOI4D (Liu et al., 2022), which often provide noisy and sparse raw depth maps, we employ Prior Depth Anything (Wang et al., 2025e) to robustly optimize them, yielding denser and more reliable depth annotations. For the public stereo dataset DROID (Khazatsky et al., 2024), we leverage FoundationStereo (Wen et al., 2025) for stereo depth estimation on this dataset.

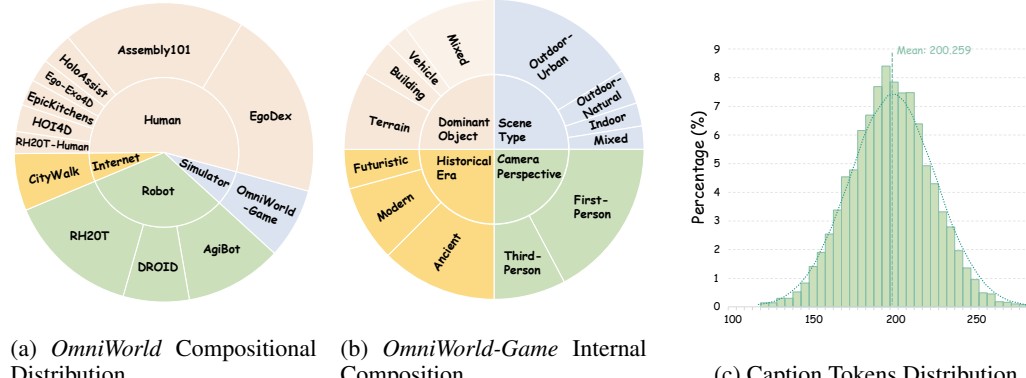

(a) *OmniWorld* Compositional Distribution

(b) *OmniWorld-Game* Internal Composition

(c) Caption Tokens Distribution

Figure 3: **Statistical information of *OmniWorld*.** (a) displays compositional distribution of data from different domains within *OmniWorld*, (b) presents internal composition of *OmniWorld-Game*. (c) shows caption tokens distribution of *OmniWorld*.

**Foreground masks.** To provide precise, temporally consistent masks of primary subjects, we develop specialized automated pipelines. For robot domain data, we use RoboEngine (Yuan et al., 2025) to generate initial masks for keyframes, followed by temporal tracking and fusion with SAM 2 (Ravi et al., 2024). For *OmniWorld-Game* (e.g., player characters in third-person view), we leverage Grounding DINO (Liu et al., 2023) to detect initial bounding boxes within predefined regions of keyframes, which then serve as prompts for SAM (Kirillov et al., 2023). These generated masks can be used as dynamic foreground masks to guide camera pose estimation.

**Camera poses.** Accurate camera pose annotation in dynamic videos is highly challenging due to transitions, weakly textured areas, and abrupt movements that hinder traditional Structure-from-Motion methods (Rockwell et al., 2025; Li et al., 2024). Following prior work (Team et al., 2025a), we develop a robust, automated, two-stage pipeline for dynamic camera pose annotation, whose principles are validated across diverse data types.

The pipeline leverages the pre-computed foreground masks to focus on static background regions. The stages include: 1) Coarse camera pose estimation leveraging VGGT (Wang et al., 2025a) for videos without depth or DroidCalib (Hagemann et al., 2023) with depth constraints; 2) Camera pose refinement through dense point tracking (SIFT (Lowe, 2004), SuperPoint (DeTone et al., 2018) with CoTracker3 (Karaev et al., 2024)) on static regions and subsequent bundle adjustment to minimize reprojection errors, optionally enhanced by forward-backward reprojection with depth information (Chen et al., 2019).

**Text captions.** We generate text descriptions using a semi-automated approach centered on the Qwen2-VL-72B-Instruct model (Wang et al., 2024a). We employ domain-specific prompting strategies for each 81-frame video segment. For instance, in the *OmniWorld-Game* domain, we generate multi-faceted descriptions covering different viewpoints (e.g., first- and third-person), character actions, background details, and camera movements.

**Optical flow.** We generate optical flow annotations using DPFlow (Morimitsu et al., 2025) to capture dense, pixel-level motion. Unlike models that require downsampling high-resolution inputs (Teed & Deng, 2020), DPFlow processes videos at their original resolution. This makes it ideal for our high resolutions dataset.

## 2.3 DATA STATISTICS

*OmniWorld* is a large-scale dataset composed of 12 distinct datasets from four domains: simulators, robots, humans, and the internet (see Tab. 2 for a summary). It contains over 600K video sequences and 300M frames with high resolutions. The dataset is richly annotated with multiple modalities, including depth, camera poses, text, optical flow, and foreground masks.

As shown in Fig. 3a, the human domain constitutes the largest portion of *OmniWorld*, highlighting its focus on real-world activities. Our self-collected *OmniWorld-Game* subset is particularly diverse, as detailed in Fig. 3b. It spans various scene types (e.g., outdoor-urban, indoor), camera perspectives

Table 3: **Monocular depth & video depth estimation** on *OmniWorld-Game* benchmark.

| Method | Mono-Depth | | Video-Depth | | | | FPS |
|---|---|---|---|---|---|---|---|
| | scale | | scale | | scale&shift | | |
| | Abs Rel ↓ | $\delta<1.25$ ↑ | Abs Rel ↓ | $\delta<1.25$ ↑ | Abs Rel ↓ | $\delta<1.25$ ↑ | |
| DUSt3R (Wang et al., 2024c) | 0.742 | 0.460 | 0.709 | 0.447 | 0.379 | 0.560 | 0.96 |
| MASt3R (Leroy et al., 2024) | 0.485 | 0.560 | 0.482 | 0.579 | 0.217 | 0.724 | 0.79 |
| MonST3R (Zhang et al., 2024) | 0.670 | 0.493 | 0.669 | 0.505 | 0.272 | 0.648 | 0.95 |
| Fast3R (Yang et al., 2025a) | 0.755 | 0.404 | 0.741 | 0.384 | 0.464 | 0.531 | 14.99 |
| CUT3R (Wang et al., 2025b) | 0.624 | 0.518 | 0.690 | 0.479 | 0.429 | 0.603 | 10.75 |
| FLARE (Zhang et al., 2025) | 0.664 | 0.475 | 0.757 | 0.453 | 0.511 | 0.527 | 4.24 |
| VGGT (Wang et al., 2025a) | 0.531 | 0.554 | **0.440** | **0.625** | **0.194** | **0.755** | **18.75** |
| MoGe-1 (Wang et al., 2024b) | 0.459 | 0.586 | – | – | – | – | – |
| MoGe-2 (Wang et al., 2025c) | **0.401** | **0.589** | – | – | – | – | – |

(first-person and third-person), historical eras (ancient to futuristic), and dominant objects (natural terrain, architecture, vehicles). This multi-dimensional diversity ensures the data is both challenging and comprehensive.

Furthermore, *OmniWorld* features structured and detailed text annotations. The captions typically range from 150 to 250 tokens (Fig. 3c), a density that significantly surpasses other large-scale video-text datasets like OpenVid-1M (Nan et al., 2024) and Panda-70M (Chen et al., 2024).

## 3 *OmniWorld-Game* BENCHMARK

### 3.1 3D GEOMETRIC PREDICTION BENCHMARK

**Benchmark design.** Existing benchmarks for 3D Geometric Foundation Models (GFMs) often feature short sequences and limited motion dynamics. For example, Sintel (Butler et al., 2012) sequences average only 50 frames, while datasets like Bonn (Palazzolo et al., 2019) and KITTI (Geiger et al., 2013) are confined to specific scenarios (e.g., indoor human motion or outdoor street views). These limitations hinder the comprehensive evaluation of a model's long-term and complex-scene modeling capabilities. To address this, our *OmniWorld-Game* benchmark provides a more challenging testbed featuring extended, high-resolution sequences (up to 384 frames at 720P) with diverse and complex motions.

**Evaluation details.** We evaluate a suite of recent GFMs, including DUSt3R (Wang et al., 2024c), MASt3R (Leroy et al., 2024), MonST3R (Zhang et al., 2024), Fast3R (Yang et al., 2025a), CUT3R (Wang et al., 2025b), FLARE (Zhang et al., 2025), VGGT (Wang et al., 2025a), and MoGe (Wang et al., 2024b; 2025c). The evaluation is conducted on two tasks: monocular depth estimation and video depth estimation.

**Analysis.** Our evaluation on *OmniWorld-Game* reveals significant challenges for current GFMs (Tab 3). For monocular depth estimation, MoGe-2 (Wang et al., 2025c) achieves the best quantitative results. This finding is supported by qualitative results in the supplementary materials, where it produces visibly sharper depth maps. In the more demanding video depth estimation task, VGGT (Wang et al., 2025a) demonstrates superior accuracy and efficiency (FPS). Point cloud visualizations (Fig. 4) confirm that VGGT generates more coherent 3D structures, yet even it produces noticeable artifacts in highly dynamic scenes. No single GFM masters all tasks on *OmniWorld-Game*. The results highlight that current SOTA models still struggle with long-sequence consistency and complex dynamics, validating our benchmark as a challenging testbed for advancing future research.

### 3.2 CAMERA-CONTROLLED VIDEO GENERATION BENCHMARK

**Benchmark design.** Existing benchmarks for camera-controlled video generation, such as RealEstate10K (Zhou et al., 2018), are often limited to static scenes with smooth camera paths, failing to reflect real-world complexity. In contrast, our *OmniWorld-Game* benchmark provides a more challenging evaluation environment, featuring rich dynamic content, complex camera trajectories, and diverse scenes.

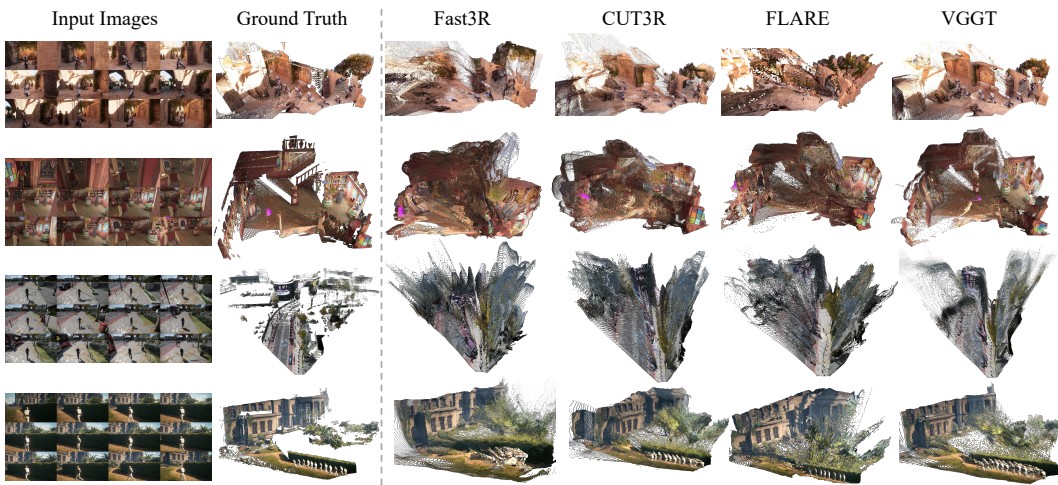

Figure 4: **Qualitative comparison of multi-view 3D reconstruction** on *OmniWorld-Game* benchmark.

Table 4: **Camera-controlled video generation evaluation** on *OmniWorld-Game* benchmark.

| Method | TransErr↓ | RotErr↓ | CamMC↓ | FVD VideoGPT↓ | FVD StyleGAN↓ |
|---|---|---|---|---|---|
| AC3D (T2V) (Bahmani et al., 2024) | 6.2788 | 0.8867 | 6.6965 | 1745.778 | 1594.885 |
| MotionCtrl (I2V) (Wang et al., 2024d) | 7.8633 | 1.1402 | 8.2710 | 694.342 | 745.652 |
| CamCtrl (I2V) (He et al., 2024) | **1.2882** | **0.2022** | **1.3856** | **615.417** | **637.574** |
| CAMI2V (I2V) (Zheng et al., 2024) | 5.9626 | 0.5087 | 6.2010 | 837.185 | 742.594 |

**Evaluation details.** We evaluate several recent models, including the Text-to-Video model AC3D (Bahmani et al., 2024) and Image-to-Video models like CamCtrl (He et al., 2024), MotionCtrl (Wang et al., 2024d), CAMI2V (Zheng et al., 2024). Following prior work (Zheng et al., 2024), we assess performance using two sets of metrics: camera-controlled accuracy (RotError, TransError, CamMC) and perceptual quality (Fréchet Video Distance, FVD) (Unterthiner et al., 2018).

**Analysis.** Our analysis on *OmniWorld-Game* reveals current models failing to achieve either high generation quality or precise camera control. For instance, the Text-to-Video model AC3D (Bahmani et al., 2024) generates subtle dynamics and fails to follow camera paths, resulting in poor quantitative and qualitative scores (Tab. 4, Fig. 5). Among Image-to-Video (I2V) models, CamCtrl (He et al., 2024) shows better quantitative performance. However, its generated videos often suffer from blurry moving characters as shown in Fig. 5. Other methods, including MotionCtrl (Wang et al., 2024d) and CAMI2V (Zheng et al., 2024), face similar quality degradation issues. These results underscore the unique challenges posed by our benchmark in evaluating spatio-temporal generation capabilities.

## 4 MODEL FINE-TUNING AND EFFICACY VALIDATION

### 4.1 IMPROVING 3D GEOMETRIC PREDICTION WITH *OmniWorld*

To demonstrate *OmniWorld*'s value as a training resource, we fine-tune three baseline models, DUSt3R (Wang et al., 2024c), CUT3R (Wang et al., 2025b), and Reloc3r (Dong et al., 2024), on subsets of our dataset. The fine-tuned models consistently surpass their original performance across monocular depth estimation (Tab. 5), video depth estimation (Tab. 6), and camera pose estimation (see supplementary). Notably, for monocular depth, the fine-tuned DUSt3R not only improved upon its baseline but also outperformed MonST3R Zhang et al. (2024), which is fine-tuned on several existing dynamic datasets. The enhancements in video depth estimation also underscore *OmniWorld*'s effectiveness in improving temporal consistency. These results validate that *OmniWorld*'s scale and diversity provide a powerful resource for boosting the generalization and robustness of 3D geometric foundation models.

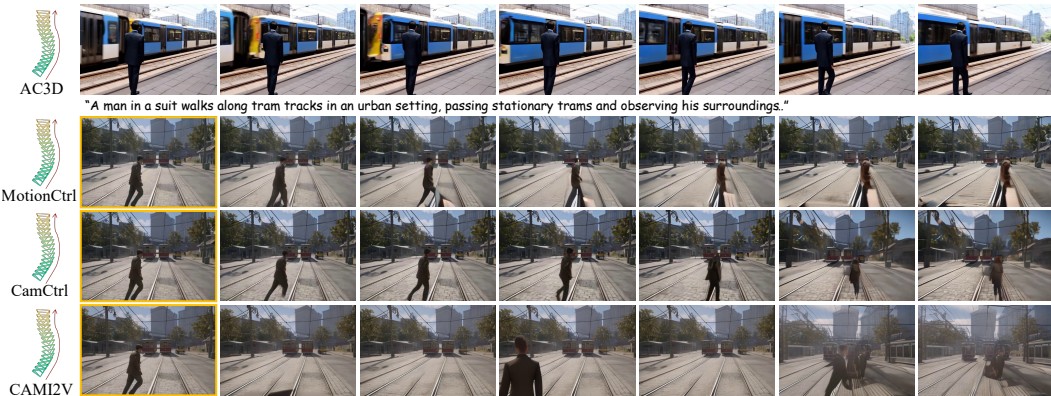

Figure 5: **Qualitative comparison of camera-controlled video generation** on *OmniWorld-Game* benchmark. AC3D takes text as a condition signal. MotionCtrl, CamCtrl, CAMI2V take an image as a condition signal. Condition images are the first images of each row.

Table 5: **Comparison of original and fine-tuned models for monocular depth estimation** on Sintel (Butler et al., 2012), Bonn (Palazzolo et al., 2019), KITTI (Geiger et al., 2013) and NYU-v2 (Silberman et al., 2012). * denotes models that have been fine-tuned on *OmniWorld*.

| Method | Sintel | | Bonn | | KITTI | | NYU-v2 | |
|---|---|---|---|---|---|---|---|---|
| | Abs Rel↓ | $\delta < 1.25$ ↑ | Abs Rel↓ | $\delta < 1.25$ ↑ | Abs Rel↓ | $\delta < 1.25$ ↑ | Abs Rel↓ | $\delta < 1.25$ ↑ |
| DUSt3R (Wang et al., 2024c) | 0.488 | **0.532** | 0.139 | 0.831 | 0.109 | 0.873 | **0.081** | **0.909** |
| MonST3R (Zhang et al., 2024) | 0.402 | 0.525 | 0.069 | **0.954** | 0.098 | 0.895 | 0.094 | 0.887 |
| DUSt3R* | **0.370** | 0.529 | **0.067** | 0.948 | 0.088 | 0.932 | 0.089 | 0.902 |
| CUT3R (Wang et al., 2025b) | 0.420 | 0.520 | **0.058** | **0.967** | 0.097 | 0.914 | 0.081 | 0.914 |
| CUT3R* | **0.408** | **0.522** | 0.075 | 0.944 | **0.087** | **0.935** | **0.075** | **0.920** |

### 4.2 ENHANCING CAMERA-CONTROLLED VIDEO GENERATION WITH *OmniWorld*

Existing datasets for camera-controlled video generation, such as RealEstate10K (Zhou et al., 2018), are often limited to static scenes with simple camera movements, which restricts a model's ability to handle dynamic content. To address this data bottleneck, we fine-tuned the AC3D (Bahmani et al., 2024) baseline on *OmniWorld*. This approach aligns with prior findings (He et al., 2025a) that highlight the critical role of dynamic data in improving camera control. As shown in Tab. 7, our fine-tuned model significantly outperforms the original baseline on both the RealEstate10K and our *OmniWorld-Game* benchmarks. This result validates *OmniWorld* as a powerful training resource for enhancing a model's capability to follow complex camera instructions in dynamic environments.

## 5 RELATED WORK

**World model dataset.** The ability of models to perform world modeling is intrinsically linked to the availability of large-scale, high-quality spatio-temporal datasets. Static 3D datasets (Dai et al., 2017; Silberman et al., 2012; Li & Snavely, 2018) have advanced 3D reconstruction by providing precise geometric information. However, their static nature limits their utility for modeling motion. In video generation, large-scale video-text datasets (Chen et al., 2024; Bain et al., 2021; Nan et al., 2024; Ju et al., 2024) offer rich semantic annotations but lack geometric information (e.g., depth), making them unsuitable for 4D world modeling. To bridge this gap, researchers have created dynamic real-world datasets for autonomous driving (Geiger et al., 2013; Sun et al., 2020) and human-robot interaction (Palazzolo et al., 2019; Liu et al., 2022; Damen et al., 2018; Fang et al., 2024). While valuable, these datasets often suffer from a lack of scene diversity and noisy geometric annotations. With advancements in modern rendering technology significantly reducing the sim-to-real gap (Wang et al., 2020), synthetic datasets have emerged as a valuable alternative providing precise annotations. However, recent synthetic datasets (Butler et al., 2012; Mayer et al., 2016; Harley et al., 2022; Wang et al., 2020; Karaev et al., 2023; Mehl et al., 2023) still fall short in terms of scale, diversity, and modal richness compared to our *OmniWorld-Game* dataset (Tab. 1).

Table 6: **Comparison of original and fine-tuned models for video depth estimation** on Sintel (Butler et al., 2012), Bonn (Palazzolo et al., 2019) and KITTI (Geiger et al., 2013). * denotes models that have been fine-tuned on *OmniWorld*.

| Method | Align | Sintel | | Bonn | | KITTI | |
|---|---|---|---|---|---|---|---|
| | | Abs Rel ↓ | $\delta < 1.25$ ↑ | Abs Rel ↓ | $\delta < 1.25$ ↑ | Abs Rel ↓ | $\delta < 1.25$ ↑ |
| DUSt3R (Wang et al., 2024c) | scale | 0.652 | 0.436 | 0.151 | 0.839 | 0.143 | **0.814** |
| DUSt3R* | | **0.512** | **0.456** | **0.083** | **0.920** | **0.135** | 0.800 |
| CUT3R (Wang et al., 2025b) | scale | 0.417 | 0.510 | 0.078 | 0.937 | 0.123 | 0.875 |
| CUT3R* | | **0.396** | **0.516** | **0.078** | **0.938** | **0.107** | **0.907** |
| DUSt3R (Wang et al., 2024c) | scale&shift | 0.570 | **0.493** | 0.152 | 0.835 | **0.135** | **0.818** |
| DUSt3R* | | **0.520** | 0.480 | **0.084** | **0.914** | 0.136 | 0.808 |
| CUT3R (Wang et al., 2025b) | scale&shift | 0.537 | 0.556 | 0.075 | 0.944 | 0.111 | 0.884 |
| CUT3R* | | **0.314** | **0.574** | **0.067** | **0.964** | **0.103** | **0.912** |

Table 7: **Comparison of original and fine-tuned models for camera-controlled video generation evaluation** on RealEstate10K (Zhou et al., 2018) and *OmniWorld-Game* benchmark. * denotes models that have been fine-tuned on *OmniWorld*.

| Method | Benchmark | TransErr↓ | RotErr↓ | CamMC↓ | FVD | |
|---|---|---|---|---|---|---|
| | | | | | VideoGPT↓ | StyleGAN↓ |
| AC3D (Bahmani et al., 2024) | RealEstate10K | 3.4433 | 0.6308 | 3.6615 | 479.320 | **409.795** |
| AC3D* | | **2.8648** | **0.5314** | **3.0518** | **472.683** | 416.948 |
| AC3D (Bahmani et al., 2024) | *OmniWorld-Game* | 6.2788 | 0.8867 | 6.6965 | 1745.778 | 1594.885 |
| AC3D* | | **4.1428** | **0.7610** | **4.4854** | **1437.247** | **1249.186** |

**3D geometric foundation models.** 3D geometric foundation models have recently emerged as a data-driven alternative to traditional methods. Early works like DUSt3R (Wang et al., 2024c) and MonST3R (Zhang et al., 2024) operate on image pairs, requiring expensive global alignment for larger scenes. Further research has introduced diverse architectural innovations to overcome this, including parallel processing (Fast3R (Yang et al., 2025a)), decomposing the learning task (FLARE (Zhang et al., 2025)), online processing for image streams (CUT3R (Wang et al., 2025b)), multi-task learning (VGGT (Wang et al., 2025a)), and permutation-equivariant designs ($\pi^3$ (Wang et al., 2025d)). However, the efficacy of these models is fundamentally tied to large-scale, multimodal training data. We validate *OmniWorld* as a powerful training resource that fulfills this need.

**Camera-controlled video generation.** Most methods in this field inject camera parameters (such as Plücker embeddings) into a pre-trained video diffusion model (Blattmann et al., 2023; Chen et al., 2023; Yang et al., 2024b) with representative works including MotionCtrl (Wang et al., 2024d), CameraCtrl (He et al., 2024), CAMI2V (Zheng et al., 2024), AC3D (Bahmani et al., 2024). Despite this progress, these methods still struggle to generate dynamic content with complex camera control. They are typically trained on datasets like RealEstate10K (Zhou et al., 2018) or DL3DV-10K (Ling et al., 2024), which consist of static scenes with smooth camera motions. This data limitation inherently restricts them to handle dynamic scenes (He et al., 2025a). The performance gap is evident on our challenging *OmniWorld-Game* benchmark.

## 6 CONCLUSION

We introduce *OmniWorld*, a large-scale, multi-domain, and multi-modal dataset designed to address the critical data bottleneck for 4D world modeling. By integrating self-collected *OmniWorld-Game* dataset and several public datasets from various domains, we create a comprehensive data resource for 4D world modeling. We demonstrate that *OmniWorld-Game* serves as a challenging benchmark for 3D geometric prediction and camera-controlled video generation, revealing the limitations of current methods. Furthermore, we provide strong evidence that fine-tuning with *OmniWorld* significantly boosts the performance of these models, underscoring its value as a powerful training resource. We believe that *OmniWorld* will serve as a crucial data resource for the community, accelerating the development of more general and robust models for understanding and interacting with the real physical world.

ACKNOWLEDGEMENTS

This work is supported by Shanghai Artificial Intelligence Laboratory.

ETHICS STATEMENT

Our work, the *OmniWorld* dataset, is a composite dataset consisting of a newly collected game-derived dataset (*OmniWorld-Game*) and several curated public datasets. We have undertaken a multi-faceted approach to ensure our practices for both collecting new data and curating existing data are legally compliant.

1) Responsible Acquisition of Game-Derived Data. Game data was captured from legally purchased games using standard, non-invasive tools (e.g., OBS, ReShade) without any reverse engineering or cheating. To respect the source material, we automatically remove UI elements and text, and manually filter for sensitive content such as story spoilers.

2) Adherence to Terms of Use for Game Content. Our use of game content is strictly non-commercial, aligning with publisher terms of service (e.g., Rockstar Games (Rockstar Games, 2024)). The dataset is intended solely to advance academic research and does not compete with or infringe upon the economic interests of the copyright holders.

3) Curation of Public Datasets. *OmniWorld* also incorporates public datasets to enhance domain diversity. We have strictly adhered to the original license of each dataset, ensuring proper attribution and compliance with all usage terms.

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

## A    OVERVIEW

This appendix provides comprehensive supplementary materials, organized as follows:

- **Dataset & Benchmarks:** Sec. B details the *OmniWorld* dataset construction. Extended details on our benchmark and fine-tuning experiments are provided in Sec. C and Sec. D, supplemented by comprehensive benchmark statistics in Sec. I.

- **Data Processing & Validation:** Sec. F details our strategy for filtering high-quality video clips. Sec. G and Sec. J provide qualitative and quantitative validations of our annotation pipelines, respectively.

- **Analysis:** Sec. H presents a comprehensive failure case analysis to discuss the current limitations.

## B    *OmniWorld* DATASET

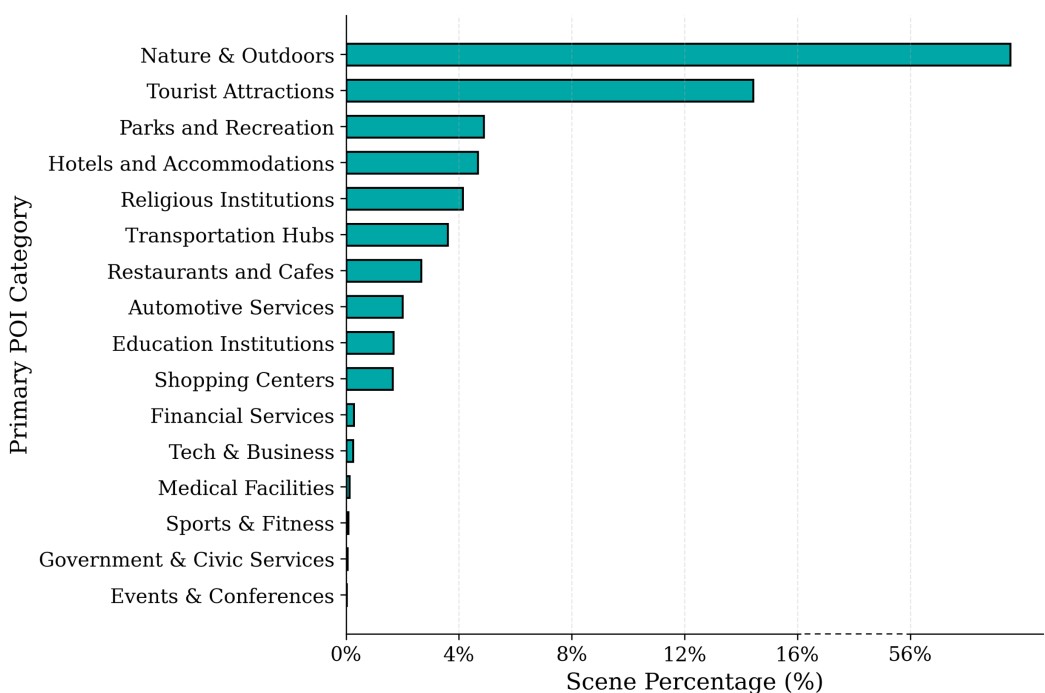

Figure 6: **The *OmniWorld-Game* distribution** of scene category (the primary POI locations).

To quantitatively analyze the scene diversity of *OmniWorld-Game*, we adopt the methodology from DL3DV (Ling et al., 2024) to classify and count scenes across 16 Point-of-Interest (POI) categories (Ye et al., 2011). The statistical results are shown in Fig. 6. *OmniWorld-Game* encompasses a wide variety of scene categories, including "Nature & Outdoors", "Tourist Attractions", "Parks and Recreation", and "Hotels and Accommodations". "Nature & Outdoors" represents the largest share, reflecting its dominant presence in the dataset. The distribution of these scene categories aligns with their prevalence in the real world and the characteristics of the games themselves. For instance, scenes related to "Government & Civic Services" and "Events & Conferences" are typically less frequent in games, leading to their lower representation in our dataset. These statistics further validate the richness and real-world attributes of *OmniWorld-Game*.

To provide a more detailed analysis of the dominant "Nature & Outdoors" scenes in *OmniWorld-Game*, we further subdivide this category into 5 second-level and 40 third-level categories. The detailed distribution is shown in Fig. 7. Our statistics reveal that "Natural Landforms & Ecosystems" is the dominant second-level category. Within this category, scenes depicting "Forests &

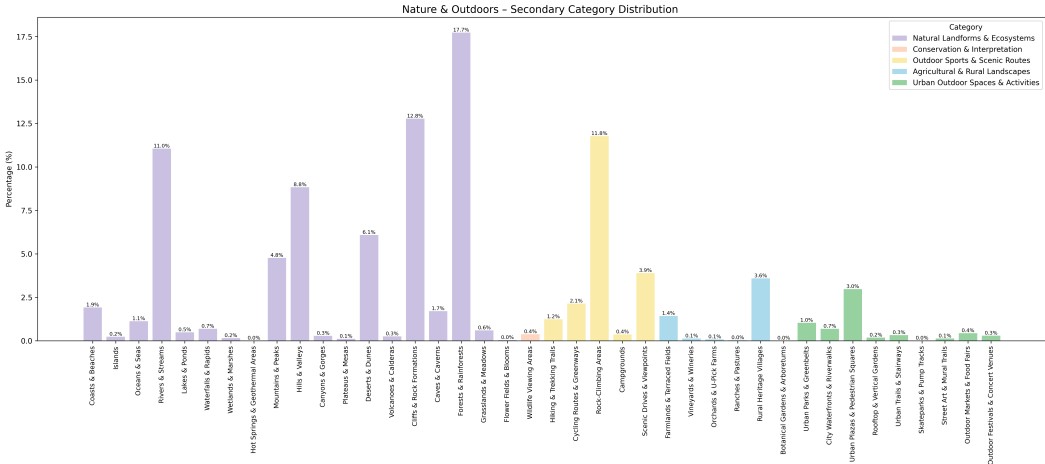

Figure 7: **Scene diversity within the "Nature & Outdoors" category.** A quantitative breakdown of second- and third-level scene categories in *OmniWorld-Game* dataset, demonstrating the high internal diversity and distribution of natural environments.

Rainforests" and "Cliffs & Rock Formations" are the most prevalent. "Outdoor Sports & Scenic Routes" is the second-largest category, with scenes of "Rock-Climbing Areas" and "Scenic Drives & Viewpoints" being particularly prominent. Additionally, "Urban Outdoor Spaces & Activities" and "Agricultural & Rural Landscapes" also make up a small portion of the data. These detailed statistics confirm that the "Nature & Outdoors" scenes in *OmniWorld-Game* are not only abundant but also internally diverse. This rich composition provides a diverse data source for world modeling in complex natural environments.

## C  *OmniWorld-Game* BENCHMARK

### C.1  3D GEOMETRIC PREDICTION

**Experiment details.** We adhere to the default configurations of each evaluated model. The entire evaluation process is conducted on a single A800 GPU. All images are consistently resized to a long side of 512 pixels while preserving aspect ratio.

For the monocular depth estimation, we evaluate the first 200 frames of 18 test sequences from the *OmniWorld-Game* benchmark. Following the evaluation protocols of prior works (Zhang et al., 2024; Wang et al., 2025b;d), we focus on scale-invariant monocular depth accuracy. The primary evaluation metrics are Absolute Relative Error (Abs Rel) and threshold accuracy ($\delta < 1.25$). Under this setting, the depth map of each frame is independently aligned with its corresponding ground truth.

For the video depth estimation, we select the first 100 frames of the same test sequence from the *OmniWorld-Game* benchmark. To ensure a fair comparison across all models, we cap the input sequence length at 100 frames, as some models (e.g., FLARE (Zhang et al., 2025)) cannot handle longer sequences without errors. Similar to the monocular depth estimation, we report Abs Rel and $\delta < 1.25$. To more comprehensively evaluate depth consistency across video sequences, we provide results under two different alignment settings: (i) scale-only alignment (scale) and (ii) combined scale and translation alignment (scale & shift). These settings test a model's depth estimation capabilities under different constraints, particularly in handling motion and viewpoint changes.

It is important to note that since the benchmark data is included in the training set of $\pi^3$ (Wang et al., 2025d), we did not evaluate it in our benchmark.

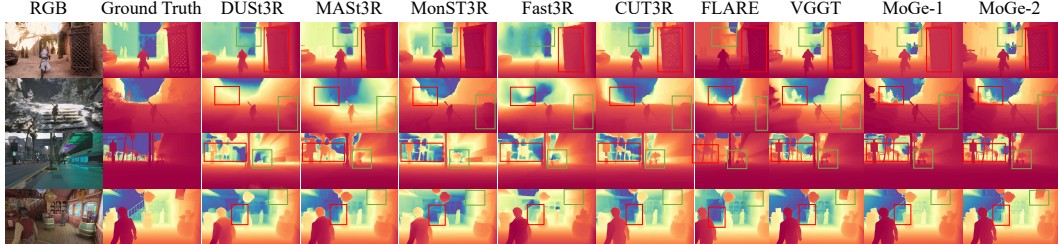

Figure 8: **Qualitative comparison of monocular depth estimation** on *OmniWorld-Game* benchmark across various methods.

Table 8: **Comparison of original and fine-tuned models for camera pose estimation** on Sintel (Butler et al., 2012), TUM-dynamics (Sturm et al., 2012) and ScanNet (Dai et al., 2017). * denotes models that have been fine-tuned on *OmniWorld*.

| Method | Sintel | | | TUM-dynamics | | | ScanNet | | |
|---|---|---|---|---|---|---|---|---|---|
| | ATE↓ | RPE trans↓ | RPE rot↓ | ATE↓ | RPE trans↓ | RPE rot↓ | ATE↓ | RPE trans↓ | RPE rot↓ |
| CUT3R (Wang et al., 2025b) | 0.210 | 0.071 | **0.627** | 0.045 | 0.014 | 0.441 | 0.096 | 0.022 | 0.733 |
| CUT3R* | **0.178** | **0.055** | 0.651 | **0.041** | **0.013** | **0.374** | **0.095** | **0.022** | **0.604** |

## C.2 CAMERA-CONTROLLED VIDEO GENERATION

**Experiment details.** AC3D (Bahmani et al., 2024) uses CogVideoX-5B (Yang et al., 2024b) as base T2V model, it generates 25 frames per inference at a resolution of $480 \times 720$. CamCtrl (He et al., 2024) and MotionCtrl (Wang et al., 2024d) use Stable Video Diffusion (SVD) (Blattmann et al., 2023) as base I2V model and generate 14-frame video sequences at a resolution of $320 \times 512$. CAMI2V (Zheng et al., 2024) uses DynamiCrafter (Xing et al., 2023) as base I2V model. It generates 16-frame video sequences at a resolution of $320 \times 512$. For a fair comparison with CamCtrl and MotionCtrl, we use the first 14 frames of its generated videos for evaluation. We use $\pi^3$ (Wang et al., 2025d) to get camera poses of the generated videos. All methods are evaluated on an A800 GPU.

## C.3 ADDITIONAL QUALITATIVE RESULTS

We provide additional qualitative results that complement the quantitative analysis in Sec. 3.1 of the main paper. Fig. 8 provides a visual comparison of monocular depth estimation results from various methods on our *OmniWorld-Game* benchmark. These visualizations confirm that MoGe-2 (Wang et al., 2025c) generates depth maps with significantly sharper details and more coherent geometric structures compared to its counterparts.

# D MODEL FINE-TUNING

## D.1 CAMERA POSE ESTIMATION.

Following prior work (Wang et al., 2025b;d), we report the Absolute Trajectory Error (ATE), Relative Pose Error for translation (RPE trans), and Relative Pose Error for rotation (RPE rot) on Sintel (Butler et al., 2012), TUM-dynamics (Sturm et al., 2012) and ScanNet (Dai et al., 2017). The results in Tab. 8 show that CUT3R's performance notably improved after fine-tuning on *OmniWorld* in camera pose estimation.

We perform relative camera pose evaluation on the DynPose-100K (Rockwell et al., 2025) and the *OmniWorld-CityWalk* test set. Following prior work (Dong et al., 2024), we assess performance with three indicators: AUC@5/10/20, which measure the area under the pose accuracy curve. This curve is based on minimum thresholds of 5, 10, and 20 degrees for rotation and translation angular errors. Reloc3r (Dong et al., 2024) demonstrated substantial improvements in its ability to estimate dynamic camera poses after fine-tuning on *OmniWorld* in relative camera pose evaluation (Tab. 9).

Table 9: **Comparison of original and fine-tuned models for relative camera pose evaluation** on DynPose-100K (Rockwell et al., 2025), *OmniWorld-CityWalk* (Li et al., 2025). * denotes models that have been fine-tuned on *OmniWorld*.

| Method | DynPose-100K | | | OmniWorld-CityWalk | | |
|---|---|---|---|---|---|---|
| | AUC@5↑ | AUC@10↑ | AUC@20↑ | AUC@5↑ | AUC@10↑ | AUC@20↑ |
| Reloc3r (Dong et al., 2024) | 6.9 | 15.4 | 27.1 | 33.3 | 49.4 | 63.1 |
| Reloc3r* | **14.4** | **25.5** | **37.8** | **42.5** | **58.0** | **70.3** |

## D.2 IMPLEMENTATION DETAILS

We conduct comprehensive fine-tuning experiments on several SOTAs to validate the efficacy of our *OmniWorld* as a training resource. All experiments are performed on 8 A800 GPUs.

**DUSt3R (Wang et al., 2024c).** For fine-tuning, we use *OmniWorld-Game* alongside a portion of DUSt3R's original training sets, including ARKitScenes (Baruch et al., 2021), MegaDepth (Li & Snavely, 2018), and Waymo (Sun et al., 2020). We load the pre-trained weights of DUSt3R and performed full fine-tuning. The model is fine-tuned on images with random resolutions (e.g., 288×512, 384×512, 336×512). The training runs for 40 epochs, with each epoch consisting of 800 iterations. We use the AdamW optimizer with an initial learning rate of $2.5 \times 10^{-5}$ and a weight decay of 0.05. Each GPU had a batch size of 7, with each batch containing two images.

**CUT3R (Wang et al., 2025b).** We fine-tune CUT3R using *OmniWorld-Game* and a subset of its original training data, including CO3Dv2 (Reizenstein et al., 2021), WildRGBD (Xia et al., 2024), ARKitScenes (Baruch et al., 2021), Waymo (Sun et al., 2020), and TartanAir (Wang et al., 2020). We load the pre-trained weights and follow the training strategy from CUT3R's training stage 3. We fine-tune on higher-resolution images with varied aspect ratios, setting the maximum side to 512 pixels. The encoder is frozen, with only the decoder and heads being trained on longer sequences of 4 to 64 views. The model is fine-tuned for 2,000 iterations with a total batch size of 96 and a learning rate of $1.0 \times 10^{-6}$, optimized by AdamW with a weight decay of 0.05.

**Reloc3r (Dong et al., 2024).** For fine-tuning Reloc3r, we utilize *OmniWorld-Game*, *OmniWorld-CityWalk*, *OmniWorld-HoloAssist*, and *OmniWorld-EpicKitchens*, along with a portion of its original training sets, including CO3Dv2 (Reizenstein et al., 2021), ARKitScenes (Baruch et al., 2021), Scannet++ (Yeshwanth et al., 2023), BlendedMVS (Yao et al., 2020), and MegaDepth (Li & Snavely, 2018). We load the pre-trained weights, freeze the ViT encoder, and only update the weights for the decoder and pose regression head. Fine-tuning is performed on images of random resolutions, including $288 \times 512$, $384 \times 512$, and $336 \times 512$. The model is trained for 80 epochs, with each epoch comprising 400 iterations. We use the AdamW optimizer with a learning rate of $5.0 \times 10^{-6}$ and a weight decay of 0.05. Each GPU has a batch size of 32, with each batch containing two images.

**AC3D (Bahmani et al., 2024).** We fine-tune AC3D using *OmniWorld-Game*, *OmniWorld-EpicKitchens*, *OmniWorld-HOI4D*, *OmniWorld-HoloAssist*, *OmniWorld-EgoExo4D*, and *OmniWorld-EgoDex*, as well as the original training set, RealEstate10K (Zhou et al., 2018). We load the pre-trained weights of the AC3D ControlNet (Zhang et al., 2023), which is based on CogVideoX-5B (Yang et al., 2024b). Only the ControlNet model is fine-tuned, with other network structures frozen. The fine-tuning is performed on video clips of 49 frames with a resolution of 352 $\times$ 640. The model is fine-tuned for 6,000 iterations with a total batch size of 8 and a learning rate of $5.0 \times 10^{-5}$, optimized by AdamW with a weight decay of 0.0001. The fine-tuned and original models are evaluated on two distinct benchmarks: a random subset of 150 video samples from the RealEstate10K (Zhou et al., 2018) test set and *OmniWorld-Game* benchmark, which consists of 200 video samples. For a fair comparison, all models are configured to output videos at a uniform resolution of $720 \times 480$ with a sequence length of 25 frames.

## D.3 VISUAL RESULTS.

Fig. 9 provides a qualitative comparison of DUSt3R (Wang et al., 2024c) and CUT3R (Wang et al., 2025b) on the Sintel (Butler et al., 2012) subset of the Video Depth Estimation benchmark, evaluated both before and after fine-tuning on *OmniWorld*. After fine-tuning, both models recover finer

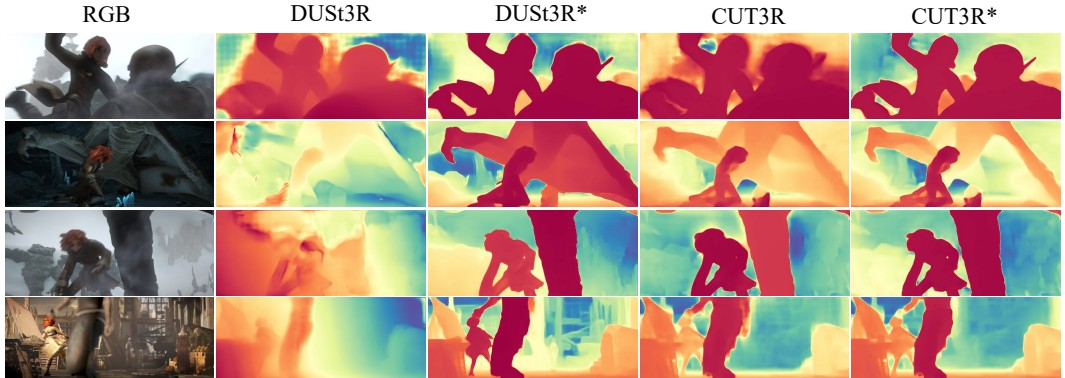

Figure 9: **Qualitative comparison of original and fine-tuned models for video depth estimation** on the Sintel (Butler et al., 2012). * denotes models that have been fine-tuned on *OmniWorld*. After fine-tuning, both models recover finer geometric details and produce more accurate depth maps, highlighting the efficacy of *OmniWorld* as a geometric supervision source.

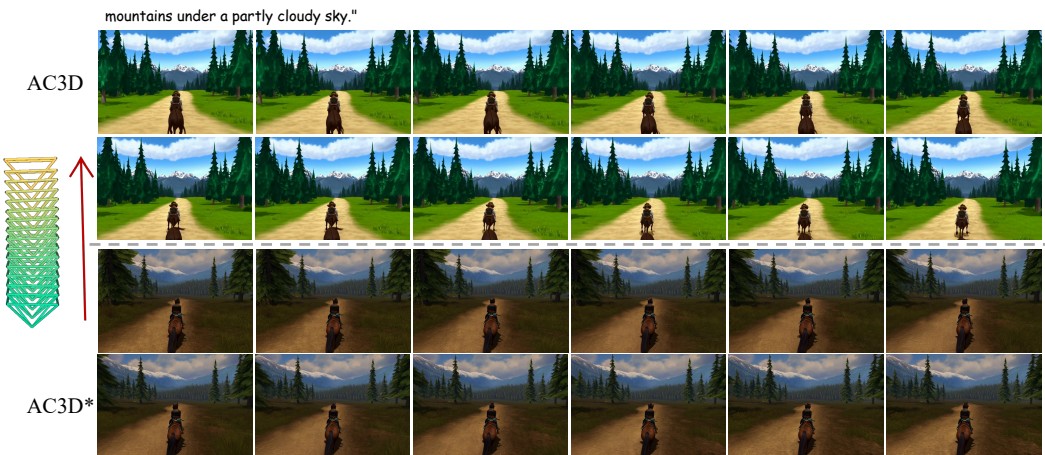

Figure 10: **Qualitative comparison of original and fine-tuned models for camera-controlled video generation.** * denotes models that have been fine-tuned on *OmniWorld*. The visualizations show that fine-tuning with our dataset significantly improves the model's ability to generate videos that more accurately follow camera trajectories and maintain higher temporal consistency for moving objects.

geometric details and generate more accurate depth maps. These results indicate that *OmniWorld* offers strong geometric supervision and can substantially enhance a model's geometric prediction capability.

Fig. 10 presents a visual comparison of AC3D (Bahmani et al., 2024) on the *OmniWorld-Game* benchmark before and after fine-tuning on the *OmniWorld* dataset for the camera-controlled video generation task. The visualizations clearly show that after fine-tuning, the generated videos more closely follow the desired camera trajectory and exhibit higher temporal consistency for moving objects. This demonstrates that *OmniWorld* can significantly enhance a model's ability to model dynamics.

# E    STATEMENT ON LLM USAGE

In the preparation of this manuscript, we use Large Language Models (LLMs) only to polish writing.

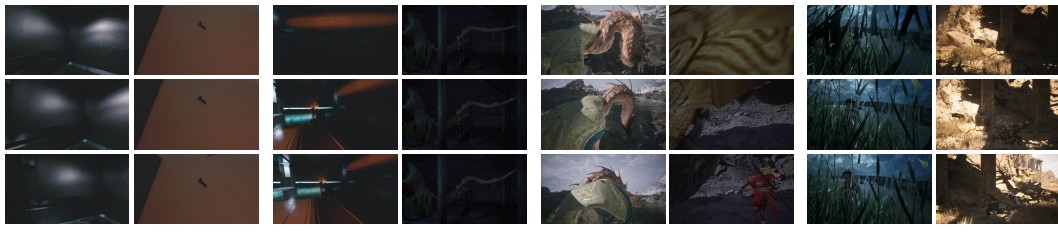

| Texture Scarcity | Severe Underexposure | Dynamic Occlusion | Motion Blur |

Figure 11: **Examples of low-quality frames discarded by the filtering pipeline.** Each column illustrates a specific failure case: Texture Scarcity, Severe Underexposure, Dynamic Occlusion, Motion Blur.

## F  HIGH-QUALITY VIDEO CLIP SELECTION

To ensure the precision and geometric consistency of camera pose annotations within the *OmniWorld* dataset, we implemented a sophisticated, multi-stage automated filtering pipeline. This pipeline is designed to extract high-quality, texture-rich, and smoothly moving video segments (Splits) from raw, long video sequences, while discarding frames unsuitable for robust 3D reconstruction. The filtering process encompasses two primary stages: frame-level quality assessment and temporal consistency validation.

### F.1  FRAME-LEVEL QUALITY ASSESSMENT

Before sequence segmentation, each individual frame undergoes a rigorous evaluation. Any frame failing to meet the following criteria is designated as an "invalid frame," triggering a truncation of the current video segment.

**Texture Scarcity**. Robust feature matching is fundamental for accurate camera pose estimation. We employ the SIFT (Lowe, 2004) algorithm for feature point extraction. A frame is discarded if it yields few valid keypoints, indicative of homogenous surfaces or extreme blur.

**Geometric Invalidity**. For source data including depth information, we check the integrity of the depth maps. A frame is deemed geometrically invalid if it contains invalid values or if the area of invalid depth (zero values) exceeds 60% of the total image pixels.

**Severe Underexposure**. Extremely dark scenes significantly degrade the signal-to-noise ratio, adversely affecting reconstruction quality. We quantify the proportion of pixels with values below 20 in the RGB image. If dark pixels constitute over a specific threshold of the frame, it is marked as unusable.

**Dynamic Occlusion**. Large-area dynamic foreground objects can confound SfM algorithms, which typically assume a static scene. Leveraging semantic masks generated by Grounding DINO (Liu et al., 2023) and SAM2 (Ravi et al., 2024), we calculate the screen occupancy of dynamic entities (e.g., characters, vehicles). If the dynamic region exceeds 60% of the frame, it is consequently excluded.

### F.2  TEMPORAL CONSISTENCY AND MOTION FILTERING

To guarantee the continuity of inter-frame relationships within video segments, we integrate optical flow estimation, utilizing RAFT (Teed & Deng, 2020), to impose constraints on adjacent frame motion characteristics.

**Forward-Backward Flow Consistency**. We compute bidirectional optical flow between frames at time $t$ and $t + 1$, and employ the forward-backward error to detect occlusions and matching inaccuracies. If the proportion of pixels satisfying geometric consistency constraints falls below 50%, it indicates a sudden scene change or severe occlusion, leading to the truncation of the current sequence.

**Motion Magnitude Limitation**. Excessive camera motion often results in motion blur and reduced inter-frame overlap, which can severely impede feature tracking. We calculate the mean magnitude of normalized optical flow. If the average motion exceeds 10% of the image dimensions (a threshold set at 0.1), it is classified as rapid motion, and the current segment is truncated. This criterion is crucial for preventing tracking loss during SfM.

Fig. 11 provides visual examples of various "bad cases" that our filtering pipeline effectively identifies and discards. These examples illustrate frames affected by texture scarcity, severe underexposure, dynamic occlusion, and significant motion blur, all of which compromise the quality of camera pose estimation.

Through this stringent filtering methodology, we effectively eliminate low-quality data, thereby ensuring that each selected video segment within *OmniWorld* is suitable for generating high-precision camera pose annotations.

## G  QUALITATIVE VALIDATION OF ANNOTATION PIPELINES

To visually substantiate the effectiveness of our annotation pipelines, we provide qualitative validation.

### G.1  CAMERA POSE ESTIMATION

We compare our two-stage pipeline (DroidCalib (Hagemann et al., 2023) initialized, refined via Co-Tracker3 (Karaev et al., 2024) and Bundle Adjustment) against the baseline DroidCalib (Hagemann et al., 2023). To visualize the accuracy of the estimated camera poses, we project the ground truth depth maps into a unified 3D point cloud using the estimated camera extrinsic parameters. As shown in Fig. 12, the baseline DroidCalib often suffers from drift or misalignment in dynamic scenes, resulting in "ghosting" artifacts or structural inconsistencies (highlighted by red arrows). In contrast, our pipeline produces globally consistent point clouds where static background geometries (e.g., walls, pillars, trees) are sharply aligned, demonstrating superior pose accuracy in the presence of dynamic foregrounds.

### G.2  FOREGROUND MASK GENERATION IN DYNAMIC SCENARIOS

We compare our semantic segmentation pipeline (Grounding DINO (Liu et al., 2023) with SAM (Ravi et al., 2024)) against the recent SOTA video-specific segmentation model SegAnyMo (Huang et al., 2025).

As illustrated in Fig. 13, our pipeline demonstrates superior mask results and robustness in complex dynamic environments. In the second row (narrow alley), our method successfully segments multiple walking pedestrians that are missed by the baseline. In the third row (road scene), SegAnyMo incorrectly segments static traffic signs as dynamic foregrounds, whereas our method correctly identifies and segments the moving vehicles. These results validate our choice of the Grounding DINO with SAM pipeline for generating reliable foreground masks, which are crucial for filtering dynamic interference in downstream tasks.

## H  FAILURE CASE ANALYSIS

While our automated pipeline (Grounding DINO (Liu et al., 2023) with SAM (Ravi et al., 2024)) demonstrates high reliability across diverse domains, we provide a transparent analysis of typical failure cases to highlight potential limitations. We identify two primary scenarios where the pipeline may exhibit imperfections: extreme crowd density and color ambiguity in close-range interactions.

As shown in the top row of Fig. 14, in highly cluttered outdoor scenes with numerous dynamic agents, the pipeline successfully segments the majority of pedestrians. However, due to resolution limitations and severe occlusion, detection may fail for distal subjects (i.e., small figures in the far background).

In ego-centric robotic manipulation scenarios (bottom row of Fig. 14), the camera often operates in close proximity to objects. When the robotic arm interacts with objects that share similar color

| Reference Image | DroidCalib | Ours |
|---|---|---|

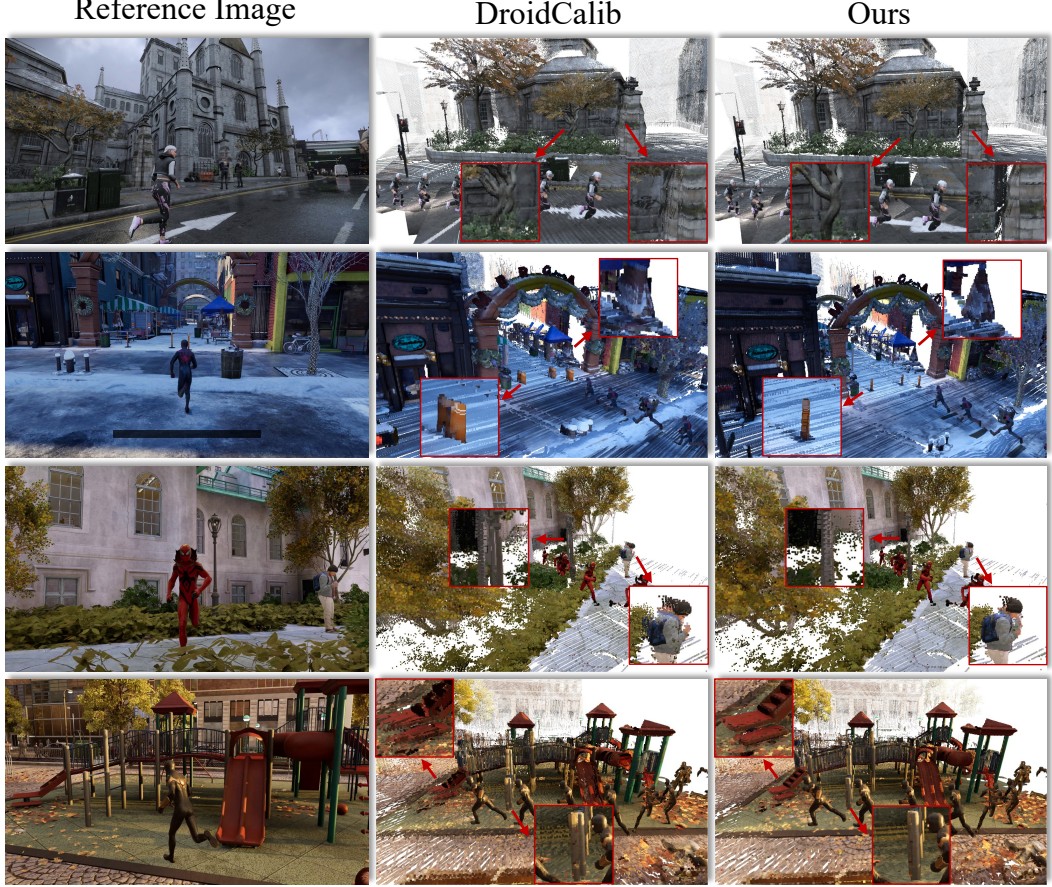

Figure 12: **Qualitative comparison of point cloud reconstructions between DroidCalib (Hagemann et al., 2023) and our camera pose estimation pipeline.** We visualize the consistency of camera poses by accumulating point clouds over a video sequence. The left column shows the reference frame. The middle column shows the reconstruction using poses from the baseline DroidCalib, where red arrows indicate significant misalignment and ghosting artifacts on static structures. The right column shows the reconstruction using ours, which effectively resolves these artifacts, resulting in sharper and more geometrically consistent 3D structures.

textures or are spatially adjacent, the segmentation mask may exhibit semantic leakage, inadvertently covering nearby static objects along with the moving arm.

Crucially, empirical observations suggest that these localized annotation imperfections have a negligible impact on our primary downstream task: dynamic camera pose estimation. As long as the dominant dynamic foregrounds are masked and a sufficient portion of the static background remains visible, the pose estimation remains accurate and stable.

# I    DETAILED BENCHMARK STATISTICS AND ANALYSIS

## I.1    GEOMETRIC PREDICTION BENCHMARK

To ensure a comprehensive evaluation of SOTA models, we employed an "Attribute-Balanced Sampling" strategy for the *OmniWorld-Game* benchmark. This section details the quantitative distributions and qualitative definitions of our test samples.

We curated a set of 90 long-sequence samples, each containing 384 frames, specifically designed to evaluate long-term geometric consistency. Each sample lasts for 16 seconds.

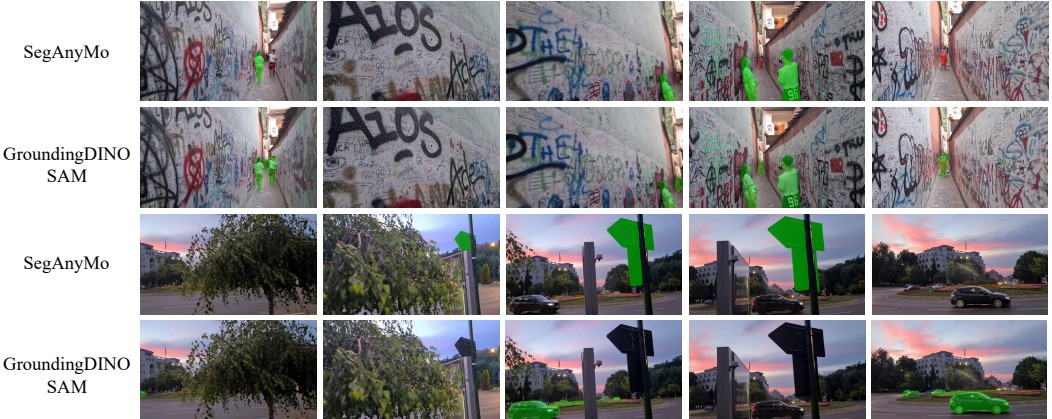

Figure 13: **Qualitative comparison of foreground mask generation on in-the-wild videos.** We compare SegAnyMo (Huang et al., 2025) with our pipeline. SegAnyMo struggles to consistently track and segment all moving subjects, while our method provides precise masks for the moving subjects.

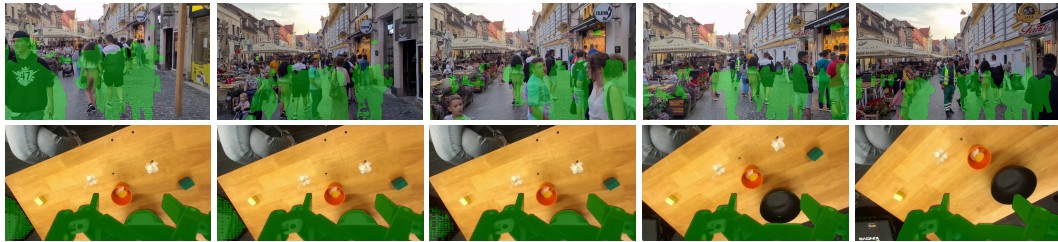

Figure 14: **Failure case analysis of foreground mask annotations.** We visualize the segmentation masks (green overlay) in challenging scenarios.

Scene Distribution. As illustrated in Fig. 15a, the dataset maintains a balance across Outdoor-Urban (58%), Outdoor-Natural (18%), and Indoor (24%) environments. This distribution ensures that models are rigorously tested on both unbounded scenes with complex backgrounds and bounded scenes featuring intricate internal structures.

Dynamic Complexity. We categorize motion into three distinct levels based on the intensity of object and camera movement. High-Dynamic (48%): Features intense motions such as running, flying, or rapid vehicle movement. Medium-Dynamic (38%): Includes standard motion (walking, running) and regular interactions. Low-Dynamic (14%): Primarily consists of background environmental motion or still character animations.

### I.2 CAMERA-CONTROLLED VIDEO GENERATION BENCHMARK

For the camera-controlled video generation task, we selected 200 samples, prioritizing trajectory complexity and environmental richness to ensure a challenging evaluation.

Camera Trajectory Complexity. As depicted in Fig. 15b, we split the samples based on camera motion. High-Complexity (51%): Features rapid translations combined with rotations, or compound high-speed movements. Medium-Complexity (49%): Represents distinct but stable motions, typical of cinematic tracking shots.

Environmental Diversity. The benchmark spans a wide range of lighting conditions (Day 60%, Night 31%, Dusk 9%) and weather scenarios (Sunny 53%, Cloudy 25%, Rain/Snow 22%).

Perspective. We include both First-person (64%) and Third-person (36%) views to test generation robustness across different field-of-view dynamics.

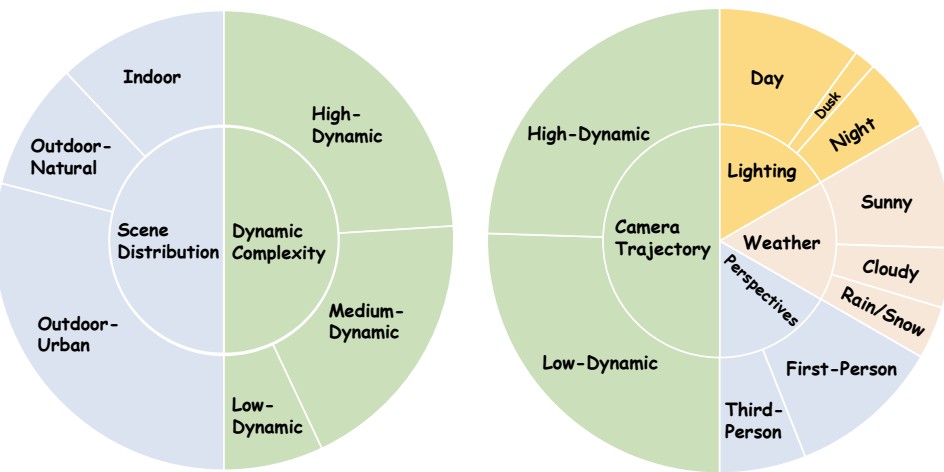

(a) Geometric Prediction Benchmark       (b) Video Generation Benchmark

Figure 15: **Statistical distribution of the *OmniWorld-Game* benchmark samples.** (a) The geometric prediction benchmark is balanced across scene types (Indoor, Outdoor) and dynamic complexity levels. (b) The video generation benchmark covers a diverse range of camera trajectory complexities, environmental conditions, and perspectives.

As shown in the visual examples in Fig. 16, *OmniWorld-Game* high-dynamic sequences introduce complex camera trajectories and drastic geometric changes, posing a substantial challenge for both geometric prediction and camera-controlled video generation.

## J   QUANTITATIVE VALIDATION OF ANNOTATION PIPELINES

To ensure the trustworthiness of the *OmniWorld* dataset, we conducted extensive quantitative analyses in key modalities.

### J.1   DEPTH ANNOTATION VALIDATION

The accuracy of our ground truth (GT) depth in *OmniWorld-Game* is intrinsically guaranteed by the rendering engine and validated by the fine-tuning experiments in Sec. 4.1. Here, we focus on validating the quality of our pseudo-labeled depth for the DROID (Khazatsky et al., 2024) dataset, which was generated using FoundationStereo (Wen et al., 2025).

To assess the utility of these annotations in downstream tasks, we pre-trained the FP3 (Yang et al., 2025b) model on point clouds projected from two sources: (1) the original DROID depth, and (2) our refined depth annotations. We then evaluated these models on four real-world tasks.

As presented in Tab. 10, the model pre-trained on our annotated depth yields significantly higher success rates across all tasks compared to the baseline using original DROID depth. This result demonstrates that our depth annotations preserve better geometric consistency.

Table 10: **Real-world manipulation task success rates (%).** Comparison of the FP3 pre-trained on the original DROID depth and our refined depth annotations. Our annotations consistently lead to higher success rates.

| Method | Open Drawer | Stack Cups | Pick up Toy | Put Toy into Basket |
|---|---|---|---|---|
| FP3 (Original DROID depth) | 25 | 10 | 60 | 35 |
| **FP3 (Ours refined depth)** | **40** | **35** | **90** | **55** |

## J.2 CAMERA POSE ANNOTATION VALIDATION

We evaluate our camera pose annotation pipeline in two scenarios: datasets without GT depth (e.g., in-the-wild videos) and datasets with GT depth (e.g., *OmniWorld-Game*).

**Scenario 1: Evaluation on Data without GT Depth.** We compared our full pipeline (VGGT (Wang et al., 2025a) initialization followed by CoTracker3 (Karaev et al., 2024) and Bundle Adjustment) against the baseline VGGT on the Sintel benchmark (Butler et al., 2012). As shown in Tab. 11, our method significantly outperforms the baseline, reducing the Absolute Trajectory Error (ATE) by over 50%. This confirms that our optimization strategy effectively refines coarse initializations into precise trajectories.

Table 11: **Pose estimation performance on the Sintel benchmark.** Comparison between the baseline VGGT and our full annotation pipeline.

| Method | ATE ↓ | RPE trans ↓ | RPE rot ↓ |
|---|---|---|---|
| VGGT (Baseline) | 0.167 | 0.062 | 0.491 |
| **Ours** | **0.082** | **0.042** | **0.246** |

**Scenario 2: Evaluation on Data with GT Depth.** To evaluate the reliability of our camera pose annotations where GT depth is available, we adopted the rigorous validation protocol following (Rockwell et al., 2025). We conducted a large-scale evaluation across 8,345 randomly sampled frame pairs from our dataset.

For each pair, we extracted high-quality sparse correspondences on static regions using Super-Point (DeTone et al., 2018) and LightGlue (Lindenberger et al., 2023), explicitly masking out dynamic objects to ensure geometric validity. We then computed the geometric consistency (via Sampson error) for poses estimated by our pipeline compared to the DroidCalib (Hagemann et al., 2023) baseline.

The results, summarized in Tab. 12, demonstrate the superiority of our approach. Our pipeline reduces the mean reprojection error to 1.09 px (from 1.30 px) and improves fine-grained precision, with 78.36% of correspondences across frame pairs falling within a 1-pixel error threshold, compared to 69.85% for the baseline. Beyond numerical metrics, our pipeline produces visibly more consistent point cloud reconstructions and the visualizations in Appendix G.1 have already demonstrated this.

Table 12: **Evaluation of camera pose annotation reliability.** We report the geometric consistency (reprojection error) on 8,345 sampled pairs, comparing our pipeline against the DroidCalib baseline.

| Method | Mean Error (px) ↓ | % < 1 Pix ↑ | % < 3 Pix ↑ | % < 5 Pix ↑ |
|---|---|---|---|---|
| DroidCalib (Baseline) | 1.30 | 69.85% | 91.42% | 96.02% |
| **Ours** | **1.09** | **78.36%** | **93.90%** | **96.66%** |

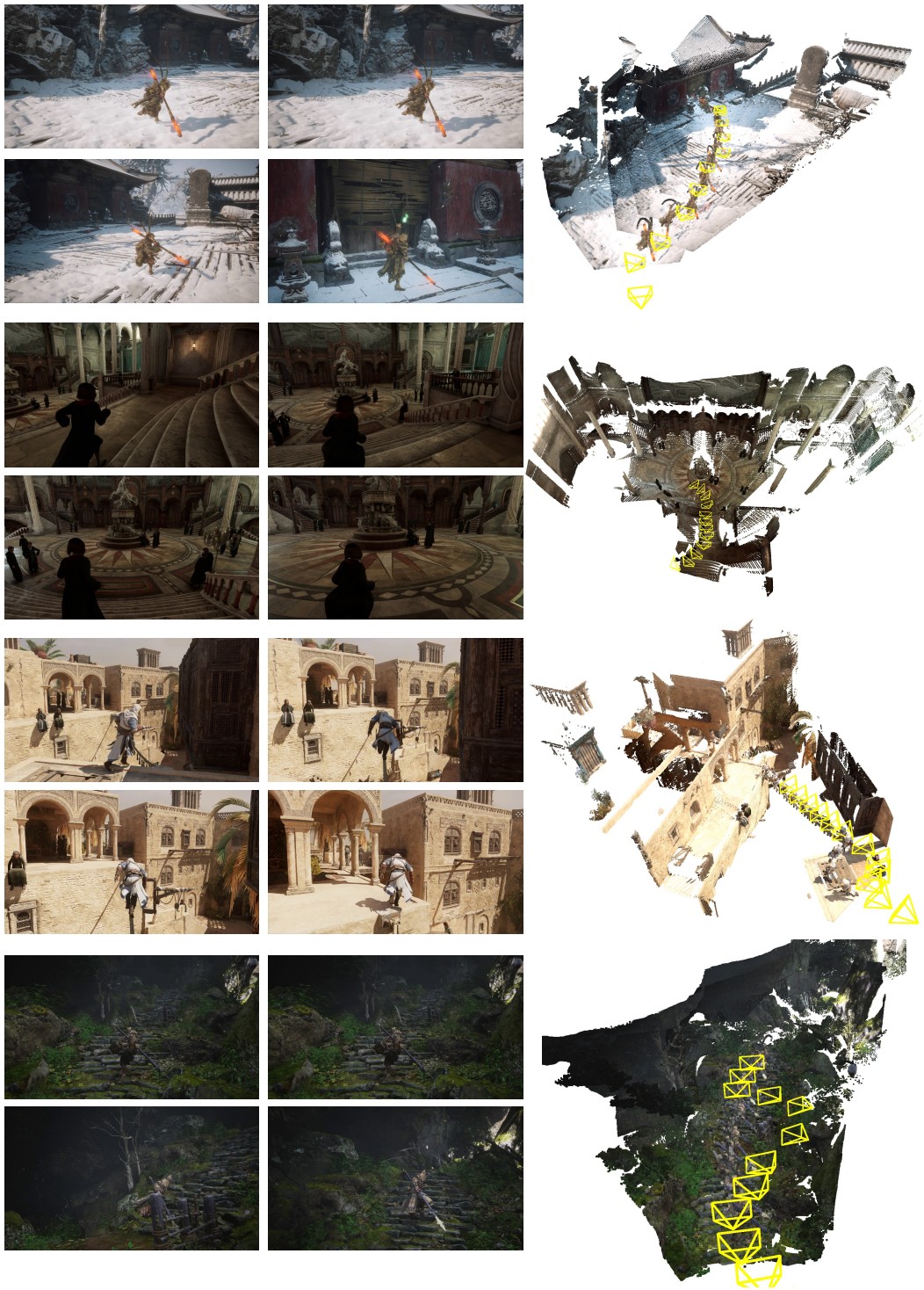

Figure 16: **Visual examples of the *OmniWorld-Game* Benchmark.** Each row displays sample RGB frames from a long sequence alongside the corresponding ground truth 3D point cloud and camera trajectory (indicated by yellow frustums). The samples demonstrate high diversity in scene types and complex camera motions, reflecting the challenging nature of the benchmark.

