# OpenReview forum: "OmniWorld: A Multi-Domain and Multi-Modal Dataset for 4D World Modeling"
_ICLR.cc/2026/Conference — ICLR 2026 Poster_

### Official Review · Reviewer_xQoM · 2025-10-28

**Soundness:** 3
**Presentation:** 3
**Contribution:** 3
**Rating:** 6
**Confidence:** 4

**Summary:**

This paper proposes OmniWorld, a large-scale dataset designed to address the lack of high-quality data for 4D world modeling.
This paper proposes OmniWorld-Game, a benchmark for 3D geometry prediction and camera control video generation.
Experiments show that fine-tuning existing state-of-the-art models on OmniWorld significantly improves their performance in 4D reconstruction and video generation tasks, validating the value of this dataset.

**Strengths:**

OmniWorld covers four major areas: simulators, robotics, humans, and the internet, and contains over 600,000 video sequences and 300 million frames of data. OmniWorld-Game significantly surpasses existing synthetic datasets in modal diversity (such as depth maps and optical flow) and data volume, providing a rich resource for 4D modeling.

Challenging benchmarks are constructed for 3D geometry prediction and camera control video generation. Multi-model evaluation reveals current state-of-the-art shortcomings. Fine-tuning experiments demonstrate that the dataset effectively improves model performance and verifies the integrity of the logic.

**Weaknesses:**

Human-domain data accounts for the largest proportion in OmniWorld, while data from other domains (such as robotics and the internet) is relatively scarce. This may lead to insufficient training of the model's generalization ability in non-human-related scenarios.

The core subset, OmniWorld-Game, is derived from a gaming environment. While it strives to simulate real-world scenarios, differences exist between game rendering and the real physical world, potentially affecting the model's transferability to real-world scenarios. Furthermore, the impact of this "simulation-to-reality" gap on task performance has not been thoroughly analyzed.

In the evaluation of 3D geometry prediction and camera control video generation, the main reliance is on quantitative indicators (such as Abs Rel, FVD) and some qualitative comparisons. There is a lack of evaluation of the model's performance in extreme scenarios (such as fast motion and complex occlusion), which cannot fully reflect the model's robustness.

**Questions:**

Regarding the problem of uneven data distribution, are there any plans to supplement data from fields such as robotics and the Internet, or use data augmentation technology to balance the data proportions in each field to improve the model's generalization capabilities in multiple scenarios?

OmniWorld-Game is derived from a game environment. How can we quantify the difference between this synthetic data and real-world data? Are there any experiments validating the performance of models trained on OmniWorld when transferred to real-world datasets (such as real indoor/outdoor 4D datasets)?

The current benchmark does not involve the evaluation of extreme scenarios (such as intense motion, severe occlusion, and low light). Will you consider expanding the benchmark's scene coverage in the future to more comprehensively measure the model's 4D modeling capabilities under complex real-world conditions?

---

> ### Author Response · Authors · 2025-11-21
>
> ### **W1: Concerns regarding data imbalance (dominance of Human domain) and potential insufficient training for non-human scenarios.**
>
> **1. OmniWorld-Game is a Massive, Standalone Foundation.** Our primary contribution is the self-collected OmniWorld-Game, which is designed to be sufficient for robust model training on its own. As shown in Table 1, it is the largest synthetic dataset currently available with comprehensive annotations for depth, camera pose, text, optical flow, and foreground masks. With over 18.5 million frames, its scale and diversity are vast enough to independently support the training of large-scale 3D Geometry Prediction models (e.g., DUSt3R, VGGT) or Video Generation models, regardless of the size of other sub-domains.
>
> **2. Augmenting Public Datasets with Critical Modalities.** While the Robot and Internet subsets are proportionally smaller than the Human domain, our contribution lies in upgrading these existing public datasets. We did not merely collect them; we enriched them by annotating missing high-value modalities (e.g., consistent depth, optical flow) that were previously unavailable. This enhancement makes these subsets significantly more valuable for multi-modal learning than their original versions.
>
> **3. Independent Utility and Future Expansion.** Each domain subset within OmniWorld is curated to be large and high-quality enough to be used as a valid sub-domain resource independently. We provide a convenient platform for the community to leverage these enriched data sources. Furthermore, OmniWorld is designed as an evolving project; we are committed to continuously integrating more data from underrepresented fields (Robotics, Internet) in future updates to further optimize the domain balance.

---

> ### Author Response · Authors · 2025-11-21
>
> ### **W2: Concerns about the potential "Sim-to-Real Gap" of OmniWorld-Game.**
>
> **We have provided extensive experiments in Sec. 4 that directly address the "Sim-to-Real" gap.**
> Our core contribution, OmniWorld-Game, surpasses existing synthetic datasets in scale and modality diversity (Table 1). As demonstrated by prior works [1], a Sim-to-Real gap can be effectively bridged with sufficiently accurate and diverse synthetic data.
>
> Our experiments validate this: as shown in **Tables 5, 6, and 8** below, fine-tuning models like DUSt3R and CUT3R **solely** on our synthetic OmniWorld-Game data leads to significant performance gains on multiple real-world benchmarks (e.g., KITTI, Bonn, NYU-v2, ScanNet) across monocular depth, video depth, and camera pose estimation tasks. This strongly quantifies and validates our synthetic data's quality and its ability to enhance real-world generalization.
>
> **Table 5: Monocular Depth Estimation.** Comparison on Real-World Benchmarks. (* denotes fine-tuned on OmniWorld-Game).
>
> | Method | Bonn: Abs Rel $\downarrow$ | Bonn: $\delta < 1.25 \uparrow$ | KITTI: Abs Rel $\downarrow$ | KITTI: $\delta < 1.25 \uparrow$ | NYU-v2: Abs Rel $\downarrow$ | NYU-v2: $\delta < 1.25 \uparrow$ |
> | :--- | :---: | :---: | :---: | :---: | :---: | :---: |
> | DUSt3R | $0.139$ | $0.831$ | $0.109$ | $0.873$ | $\mathbf{0.081}$ | $0.909$ |
> | MonST3R | $0.069$ | $\mathbf{0.954}$ | $0.098$ | $0.895$ | $0.094$ | $0.887$ |
> | **DUSt3R\*** | $\mathbf{0.067}$ | $\underline{0.948}$ | $\mathbf{0.088}$ | $\mathbf{0.932}$ | $0.089$ | $\underline{0.902}$ |
> | CUT3R | $\mathbf{0.058}$ | $\mathbf{0.967}$ | $0.097$ | $0.914$ | $0.081$ | $0.914$ |
> | **CUT3R\*** | $0.075$ | $0.944$ | $\mathbf{0.087}$ | $\mathbf{0.935}$ | $\mathbf{0.075}$ | $\mathbf{0.920}$ |
>
> **Table 6: Video Depth Estimation.** Comparison on Real-World Benchmarks. (* denotes fine-tuned on OmniWorld-Game).
>
> | Method | Align | Bonn: Abs Rel $\downarrow$ | Bonn: $\delta < 1.25 \uparrow$ | KITTI: Abs Rel $\downarrow$ | KITTI: $\delta < 1.25 \uparrow$ |
> | :--- | :---: | :---: | :---: | :---: | :---: |
> | DUSt3R | scale | $0.151$ | $0.839$ | $0.143$ | $\mathbf{0.814}$ |
> | **DUSt3R\*** | scale | $\mathbf{0.083}$ | $\mathbf{0.920}$ | $\mathbf{0.135}$ | $0.800$ |
> | CUT3R | scale | $0.078$ | $0.937$ | $0.123$ | $0.875$ |
> | **CUT3R\*** | scale | $\mathbf{0.078}$ | $\mathbf{0.938}$ | $\mathbf{0.107}$ | $\mathbf{0.907}$ |
> | DUSt3R | scale&shift | $0.152$ | $0.835$ | $\mathbf{0.135}$ | $\mathbf{0.818}$ |
> | **DUSt3R\*** | scale&shift | $\mathbf{0.084}$ | $\mathbf{0.914}$ | $0.136$ | $0.808$ |
> | CUT3R | scale&shift | $0.075$ | $0.944$ | $0.111$ | $0.884$ |
> | **CUT3R\*** | scale&shift | $\mathbf{0.067}$ | $\mathbf{0.964}$ | $\mathbf{0.103}$ | $\mathbf{0.912}$ |
>
> **Table 8: Camera Pose Estimation.** Comparison on Real-World Benchmarks. (* denotes fine-tuned on OmniWorld-Game).
>
> | Method | TUM: ATE $\downarrow$ | TUM: RPE trans $\downarrow$ | TUM: RPE rot $\downarrow$ | ScanNet: ATE $\downarrow$ | ScanNet: RPE trans $\downarrow$ | ScanNet: RPE rot $\downarrow$ |
> | :--- | :---: | :---: | :---: | :---: | :---: | :---: |
> | CUT3R | $0.045$ | $0.014$ | $0.441$ | $0.096$ | $0.022$ | $0.733$ |
> | **CUT3R\*** | $\mathbf{0.041}$ | $\mathbf{0.013}$ | $\mathbf{0.374}$ | $\mathbf{0.095}$ | $\mathbf{0.022}$ | $\mathbf{0.604}$ |
> ---
> [1] Wang et al., "TartanAir: A Dataset to Push the Limits of Visual SLAM", IROS 2020.

---

> ### Author Response · Authors · 2025-11-21
>
> ### **W3: Lack of evaluation in extreme scenarios (e.g., fast motion, complex occlusion) to fully reflect model robustness.**
>
> We appreciate this valuable insight regarding robustness. We address this by clarifying the inherent difficulty of our current benchmark and proposing a systematic categorization for finer-grained analysis.
>
> **1. OmniWorld-Game is Inherently Composed of "Extreme" Scenarios.**
> We wish to highlight that our benchmark is not comprised of simple, curated scenes; it is derived from AAA-level gameplay which inherently features the extreme conditions mentioned:
> * **Content:** It naturally contains rapid camera trajectories, severe dynamic occlusions between characters/objects, and challenging lighting conditions (e.g., caves, dusk).
> * **Evidence:** This high difficulty is precisely why current SOTA models struggle on our benchmark. As shown in **Section 3**, models exhibit significant failure cases, such as geometric artifacts in reconstruction (**Fig. 4**) and motion blur in generation (**Fig. 5**). These results confirm that our benchmark already effectively exposes model limitations under complex, non-ideal conditions.
>
> **2. Actionable Improvement: Explicit "Extreme" Subsets.**
> Constructive to your suggestion, we agree that explicitly separating these scenarios adds significant value for detailed analysis. In the final revision, we will:
> * **Systematically categorize** the test set to tag samples with specific attributes like "Fast Motion," "Heavy Occlusion," and "Low Light."
> * **Provide a fine-grained performance breakdown** on these specific "Extreme Scenario" subsets to offer a more precise and transparent measure of model robustness.

---

> ### Author Response · Authors · 2025-11-21
>
> ### **Q1: Are there plans to address uneven data distribution and improve model generalization?**
>
> We appreciate the reviewer's suggestion regarding data balance. We view OmniWorld as an evolving platform and have a clear roadmap to address this:
>
> **1. Scalability via Automated Annotation.**
> As detailed in Section 2.2, our multi-modal annotation pipeline is explicitly designed for high efficiency and scalability. This automated infrastructure removes manual bottlenecks, allowing us to rapidly process and ingest large volumes of new data from diverse sources.
>
> **2. Commitment to Future Expansion.**
> Leveraging this efficient pipeline, we are actively planning to supplement the dataset with additional data, specifically targeting the underrepresented Robotics and Internet domains. Our goal is to continuously expand OmniWorld to achieve a more balanced distribution, ensuring it provides comprehensive coverage of real-world scenarios to satisfy the research community's evolving needs for general 4D World Modeling.
>
> ### **Q2: Concerns about the potential "Sim-to-Real Gap" of OmniWorld-Game.**
>
> Please see Weakness 2.
>
> ### **Q3: Will the benchmark be expanded to include extreme scenarios for better evaluation of 4D modeling in complex conditions?**
>
> We plan to expand the benchmark. We view OmniWorld-Game as an evolving platform, and we will leverage the unique controllability of game engines to systematically capture more challenging scenarios in future updates:
>
> **1. Leveraging Engine Controllability for Environmental Extremes.**
> Unlike real-world data collection which relies on chance, we plan to utilize the rendering capabilities of game engines to actively generate "long-tail" environmental conditions. Future updates will specifically target adverse weather (e.g., heavy blizzards, sandstorms, dense fog) and challenging lighting (e.g., dynamic stroboscopic effects, extreme low-light) to rigorously test model robustness under signal degradation.
>
> **2. Targeting High-Intensity Dynamics and Occlusion.**
> To further push the boundaries of 4D modeling, we plan to expand our collection to include specific high-intensity genres. This includes high-speed aerial maneuvers for testing rapid trajectory tracking and crowd simulations for handling dynamic occlusion. These additions will ensure that OmniWorld-Game remains a challenging and comprehensive testbed as the community's modeling capabilities evolve.

---

> > ### Comment · Reviewer_xQoM · 2025-11-27
> >
> > Thank you for your reply. Your response resolved all my questions, and I have no further questions. Thank you for your solid work.

---

> > > ### Author Response · Authors · 2025-11-28
> > >
> > > We sincerely thank the reviewer for the positive feedback and for confirming that your concerns have been resolved. We appreciate your time and valuable comments, which have greatly helped improve our work.

---

### Official Review · Reviewer_XxMY · 2025-10-30

**Soundness:** 3
**Presentation:** 3
**Contribution:** 3
**Rating:** 6
**Confidence:** 3

**Summary:**

The paper introduces OmniWorld, a large-scale multi-domain and multi-modal dataset (RGB, depth, pose, text, optical flow, mask) designed for 4D world modeling. It unifies new synthetic data (OmniWorld-Game) and re-annotated public sets, supporting benchmarks for 3D geometry prediction and camera-controllable video generation. The dataset features higher resolution, richer modalities, and more complex dynamics than existing resources like Sintel or TartanAir. Its automated annotation pipeline and unified benchmark design expose clear weaknesses of current geometric and generative models, and fine-tuning on OmniWorld improves performance across multiple datasets.

However, the paper lacks clarity on how dynamic objects and camera motions are modeled, leaving its 4D claim partially unsubstantiated. The work also relies heavily on synthetic data, with limited validation on real-world domains (human, robot, internet). Moreover, the quality and uncertainty of automatically generated annotations are not quantitatively analyzed.

Overall, OmniWorld is a valuable and ambitious contribution that could serve as a strong foundation for 3D and 4D perception research. Clearer handling of dynamic scenes, stronger real-world evaluation, and annotation-quality validation would further improve the paper. My current recommendation is Borderline Accept.

**Strengths:**

1. **Significant scale and modality improvement.**
Compared with existing synthetic datasets, OmniWorld-Game offers clear advantages in resolution, frame count, and modality diversity. It also covers more dynamic and complex scenes. Table 1’s comparison against Sintel, TartanAir, and HyperSim highlights this superiority.

2. **Cross-domain integration and automated annotation pipeline.**
The paper clearly details an end-to-end annotation pipeline, from four-domain collection, video slicing and filtering, to modality annotation, and provides domain composition statistics, which support reproducibility and future extension.

3. **Challenging benchmark design.**
Compared with prior benchmarks featuring short sequences and smooth camera motion, OmniWorld-Game introduces longer sequences, complex trajectories, and dynamic scenes. On this benchmark, most state-of-the-art GFM/video generation models exhibit weaknesses in temporal consistency and camera control (as shown in Table 3/Table 4 and qualitative examples).

4. **Demonstrated training value.**
Models such as DUSt3R, CUT3R, and Reloc3R, when fine-tuned on OmniWorld, show consistent performance gains on standard public benchmarks (Sintel, KITTI, NYUv2), indicating that the dataset provides practical benefits for improving 3D geometry and temporal coherence.

**Weaknesses:**

1. **Unclear handling of dynamic 4D content.**
Given the central claim of 4D world modeling, the paper should more explicitly describe how dynamic objects and camera motions are represented during dataset construction, particularly in the synthetic (simulation) domains. It is unclear whether the dataset systematically varies object motion patterns or camera trajectories, or includes a taxonomy distinguishing static versus dynamic regions. In evaluation, the study focuses mainly on static 3D foundation models; incorporating 4D reconstruction or dynamic scene methods (e.g., MegaSAM) would make the benchmark more consistent with its 4D ambition and strengthen its empirical validity.

2. **High reliance on synthetic data and limited real-world evaluation.**
Although the authors highlight the multi-domain nature of OmniWorld, both training and evaluation are largely centered on OmniWorld-Game, the synthetic/game-based subset. Systematic experiments on the human, robot, and internet domains are limited, which makes it difficult to assess how well the dataset and benchmarks generalize to real-world conditions. A stronger cross-domain analysis would clarify the practical robustness of the proposed data.

3. **Lack of annotation-quality assessment and uncertainty analysis.**
Many modalities (depth, pose, optical flow, and foreground masks) rely on automatic or pseudo-labeling pipelines, yet the paper provides no quantitative evaluation of their accuracy or reliability. It would be valuable to include error quantification (e.g., comparisons with manual ground truth or high-confidence models), failure-case analysis, and sensitivity studies showing how annotation noise affects training and evaluation outcomes. These analyses would help validate the trustworthiness of the dataset’s supervision signals.

**Questions:**

See weaknesses.

---

> ### Author Response · Authors · 2025-11-21
>
> ### **Q1: Unclear representation of 4D dynamic content (objects/camera) and lack of explicit 4D reconstruction evaluation.**
>
> We appreciate the reviewer's careful review of our handling of 4D dynamics. We address this by clarifying our data collection strategy, our annotation taxonomy, and our expanded evaluation.
>
> **1. Data Collection. Realistic Priors and Static-Dynamic Sampling.**
> Regarding data construction, we utilized trained, experienced gamers to capture camera trajectories and object interactions that reflect realistic, human-like behavioral priors. To ensure systematic coverage, we explicitly sampled separate static and dynamic sequences in game environments that support dynamic object removal, allowing for a controlled distribution of motion patterns.
>
> **2. Taxonomy. Explicit "Static-Dynamic" Separation.**
> We employ Grounding DINO + SAM to generate high-precision Foreground Masks. These masks serve a dual purpose: providing semantic labels for moving subjects (e.g., humans, vehicles) and systematically filtering dynamic interference in our camera pose pipeline to lock onto the static background. This explicitly defines the "static background" vs. "dynamic foreground" representation.
>
> **3. Clarifying Baseline Robustness and Expanding 4D Evaluation.**
> We wish to clarify that the 3D foundation models selected (e.g., MonST3R, CUT3R, VGGT) are not limited to static scenes. They are methods specifically designed to handle geometry reconstruction "in the presence of motion", as evidenced by their original publications. Furthermore, our Camera-Controlled Video Generation task serves as a direct assessment of 4D world modeling capabilities.
>
> We fully agree with your suggestion to strengthen the "explicit 4D reconstruction" aspect. We have added an evaluation of MegaSAM, a representative optimization-based dynamic scene method, to our video depth benchmark. As shown in **Table R4**, this inclusion complements our existing feed-forward baselines and significantly betters the alignment of our benchmark with the "4D world modeling" ambition.
>
> **Table R4. Video depth estimation on OmniWorld-Game benchmark.** Comparison of scale and shift-invariant metrics. Bold indicates best, underline indicates second best.
>
> | Method | Scale: Abs Rel $\downarrow$ | Scale: $\delta < 1.25 \uparrow$ | Scale&Shift: Abs Rel $\downarrow$ | Scale&Shift: $\delta < 1.25 \uparrow$ | FPS |
> | :--- | :---: | :---: | :---: | :---: | :---: |
> | DUSt3R | $0.709$ | $0.447$ | $0.379$ | $0.560$ | $0.96$ |
> | MAST3R | $\underline{0.482}$ | $0.579$ | $\underline{0.217}$ | $\underline{0.724}$ | $0.79$ |
> | MonST3R | $0.669$ | $0.505$ | $0.272$ | $0.648$ | $0.95$ |
> | Fast3R | $0.741$ | $0.384$ | $0.464$ | $0.531$ | $\underline{14.99}$ |
> | MVDUSt3R | $1.136$ | $0.242$ | $0.825$ | $0.311$ | $0.69$ |
> | CUT3R | $0.690$ | $0.479$ | $0.429$ | $0.603$ | $10.75$ |
> | FLARE | $0.757$ | $0.453$ | $0.511$ | $0.527$ | $4.24$ |
> | VGGT | $\mathbf{0.440}$ | $\underline{0.625}$ | $\mathbf{0.194}$ | $\mathbf{0.755}$ | $\mathbf{18.75}$ |
> | MegaSAM | $0.549$ | $\mathbf{0.632}$ | $0.400$ | $0.702$ | - |

---

> ### Author Response · Authors · 2025-11-21
>
> ### **Q2: Concerns about the heavy reliance on synthetic data (OmniWorld-Game), the potential "Sim-to-Real Gap" and a lack of systematic experimental analysis on the other data domains (e.g., human, internet).**
>
> **1. We have provided extensive experiments in Sec. 4 that directly address the "Sim-to-Real" gap.**
> Our core contribution, OmniWorld-Game, surpasses existing synthetic datasets in scale and modality diversity (Table 1). As demonstrated by prior works [1], a Sim-to-Real gap can be effectively bridged with sufficiently accurate and diverse synthetic data.
>
> Our experiments validate this: as shown in **Tables 5, 6, and 8** below, fine-tuning models like DUSt3R and CUT3R **solely** on our synthetic OmniWorld-Game data leads to significant performance gains on multiple real-world benchmarks (e.g., KITTI, Bonn, NYU-v2, ScanNet) across monocular depth, video depth, and camera pose estimation tasks. This strongly quantifies and validates our synthetic data's quality and its ability to enhance real-world generalization.
>
> **Table 5: Monocular Depth Estimation.** * denotes fine-tuned on OmniWorld-Game.
>
> | Method | Bonn: Abs Rel $\downarrow$ | Bonn: $\delta < 1.25 \uparrow$ | KITTI: Abs Rel $\downarrow$ | KITTI: $\delta < 1.25 \uparrow$ | NYU-v2: Abs Rel $\downarrow$ | NYU-v2: $\delta < 1.25 \uparrow$ |
> | :--- | :---: | :---: | :---: | :---: | :---: | :---: |
> | DUSt3R | $0.139$ | $0.831$ | $0.109$ | $0.873$ | $\mathbf{0.081}$ | $0.909$ |
> | MonST3R | $0.069$ | $\mathbf{0.954}$ | $0.098$ | $0.895$ | $0.094$ | $0.887$ |
> | **DUSt3R\*** | $\mathbf{0.067}$ | $\underline{0.948}$ | $\mathbf{0.088}$ | $\mathbf{0.932}$ | $0.089$ | $\underline{0.902}$ |
> | CUT3R | $\mathbf{0.058}$ | $\mathbf{0.967}$ | $0.097$ | $0.914$ | $0.081$ | $0.914$ |
> | **CUT3R\*** | $0.075$ | $0.944$ | $\mathbf{0.087}$ | $\mathbf{0.935}$ | $\mathbf{0.075}$ | $\mathbf{0.920}$ |
>
> **Table 6: Video Depth Estimation.**
>
> | Method | Align | Bonn: Abs Rel $\downarrow$ | Bonn: $\delta < 1.25 \uparrow$ | KITTI: Abs Rel $\downarrow$ | KITTI: $\delta < 1.25 \uparrow$ |
> | :--- | :---: | :---: | :---: | :---: | :---: |
> | DUSt3R | scale | $0.151$ | $0.839$ | $0.143$ | $\mathbf{0.814}$ |
> | **DUSt3R\*** | scale | $\mathbf{0.083}$ | $\mathbf{0.920}$ | $\mathbf{0.135}$ | $0.800$ |
> | CUT3R | scale | $0.078$ | $0.937$ | $0.123$ | $0.875$ |
> | **CUT3R\*** | scale | $\mathbf{0.078}$ | $\mathbf{0.938}$ | $\mathbf{0.107}$ | $\mathbf{0.907}$ |
> | DUSt3R | scale&shift | $0.152$ | $0.835$ | $\mathbf{0.135}$ | $\mathbf{0.818}$ |
> | **DUSt3R\*** | scale&shift | $\mathbf{0.084}$ | $\mathbf{0.914}$ | $0.136$ | $0.808$ |
> | CUT3R | scale&shift | $0.075$ | $0.944$ | $0.111$ | $0.884$ |
> | **CUT3R\*** | scale&shift | $\mathbf{0.067}$ | $\mathbf{0.964}$ | $\mathbf{0.103}$ | $\mathbf{0.912}$ |
>
> **Table 8: Camera Pose Estimation.**
>
> | Method | TUM: ATE $\downarrow$ | TUM: RPE trans $\downarrow$ | TUM: RPE rot $\downarrow$ | ScanNet: ATE $\downarrow$ | ScanNet: RPE trans $\downarrow$ | ScanNet: RPE rot $\downarrow$ |
> | :--- | :---: | :---: | :---: | :---: | :---: | :---: |
> | CUT3R | $0.045$ | $0.014$ | $0.441$ | $0.096$ | $0.022$ | $0.733$ |
> | **CUT3R\*** | $\mathbf{0.041}$ | $\mathbf{0.013}$ | $\mathbf{0.374}$ | $\mathbf{0.095}$ | $\mathbf{0.022}$ | $\mathbf{0.604}$ |
>
> **2. We would like to clarify the misunderstanding regarding the lack of experiments on other domains.**
> Our experiments were not confined to synthetic data. The fine-tuning for camera pose estimation (**Table 9**) and video generation (**Table 7**) explicitly used a **multi-domain dataset**, including our annotated data from the Simulator, Human, and Internet domains (e.g., OmniWorld-Game, HOI4D, EpicKitchens, HoloAssist, EgoExo4D, EgoDex, CityWalk). The performance boosts for Reloc3r and AC3D empirically prove the effectiveness of our multi-domain data fusion.
>
> **Table 7: Camera-Controlled Video Generation.**
>
> | Method | Benchmark | TransErr $\downarrow$ | RotErr $\downarrow$ | CamMC $\downarrow$ | FVD (VideoGPT) $\downarrow$ | FVD (StyleGAN) $\downarrow$ |
> | :--- | :--- | :---: | :---: | :---: | :---: | :---: |
> | AC3D | RealEstate10K | $3.4433$ | $0.6308$ | $3.6615$ | $479.320$ | $\mathbf{409.795}$ |
> | **AC3D\*** | RealEstate10K | $\mathbf{2.8648}$ | $\mathbf{0.5314}$ | $\mathbf{3.0518}$ | $\mathbf{472.683}$ | $416.948$ |
> | AC3D | OmniWorld-Game | $6.2788$ | $0.8867$ | $6.6965$ | $1745.778$ | $1594.885$ |
> | **AC3D\*** | OmniWorld-Game | $\mathbf{4.1428}$ | $\mathbf{0.7610}$ | $\mathbf{4.4854}$ | $\mathbf{1437.247}$ | $\mathbf{1249.186}$ |
>
> **Table 9: Relative Camera Pose Evaluation.**
>
> | Method | DynPose-100K: AUC@5 $\uparrow$ | DynPose-100K: AUC@10 $\uparrow$ | OmniWorld-CityWalk: AUC@5 $\uparrow$ | OmniWorld-CityWalk: AUC@10 $\uparrow$ |
> | :--- | :---: | :---: | :---: | :---: |
> | Reloc3r | $6.9$ | $15.4$ | $33.3$ | $49.4$ |
> | **Reloc3r\*** | $\mathbf{14.4}$ | $\mathbf{25.5}$ | $\mathbf{42.5}$ | $\mathbf{58.0}$ |
> ---
> [1] Wang et al., "TartanAir: A Dataset to Push the Limits of Visual SLAM", IROS 2020.

---

> ### Author Response · Authors · 2025-11-21
>
> ### **Q3: Concerns regarding the quality of our pseudo-labels, the lack of quantitative accuracy assessment and uncertainty analysis.**
>
> We thank the reviewer for this critical question. We validate our dataset's quality through five key points: (1) a core clarification on our Ground Truth data, (2) a robust, multi-stage annotation pipeline, (3) efficacy validation via downstream task improvement, (4) multi-level filtering, and (5) new quantitative and qualitative analyses for key modalities.
>
> **1. We must clarify that our primary contribution, the OmniWorld-Game dataset, features pixel-perfect Ground Truth depth obtained directly from ReShade.**
> This is not a pseudo-label and fundamentally guarantees the accuracy of our core data. As shown in Table 1, OmniWorld-Game marks a significant leap in scale, featuring **18.5M frames with GT depth**, $18\times$ larger than classic synthetic datasets like TartanAir (1M frames), while uniquely offering dense annotations across depth, camera, text, flow, and Fg. masks. Its scale and high-precision ground truth make it an invaluable resource for the community. To the best of our knowledge, our data has already been adopted by several influential works for either evaluating or improving dynamic scene reconstruction, highlighting its effectiveness and growing influence.
>
> **2. A Customized, Multi-Stage Pipeline Ensures Quality.**
> We do not naively apply a single pre-trained model. Our pipeline (Sec 2.1, 2.2) is robustly designed:
> * **a) Strict Pre-filtering.** We apply "video slicing" to actively discard low-quality clips (e.g., motion blur, insufficient features) before annotation (Sec 2.1). For more detailed descriptions and bad case visualizations, please see the newly added **Appendix F**.
> * **b) Differentiated Annotation.** While OmniWorld-Game has GT depth, for other domains, we select SOTA models based on their specific data characteristics (e.g., stereo vs. sparse depth) and have rigorously validated their accuracy before deployment. Qualitative comparisons added to the **Appendix G** further confirm the superiority of our selected models over other methods.
> * **c) Multi-stage Optimization.** For dynamic camera pose, we employ a robust two-stage pipeline (e.g., DroidCalib/VGGT + CoTracker3 + BA). This optimization is proven to reduce errors [3].
>
> **3. Uncertainty is Handled via Multi-Level Filtering.**
> * **a) Source Filtering.** As mentioned, we filter low-quality clips pre-annotation (Sec 2.1, Appendix F).
> * **b) Post-filtering.** We will **open source** our data and dataloaders with built-in filtering rules (e.g., focal length range, depth thresholds, forward-backward optical flow consistency) to ensure users access clean data. Moreover, we added a "Failure Case Analysis" section to the **Appendix H** to transparently discuss limitations (e.g., mask inaccuracies on lots of moving objects).
>
> **4. Downstream Performance Gains Validate Data Quality.**
> Following standard practice for large-scale datasets [1, 2], the most direct validation is the performance gain on downstream tasks. As shown in Sec. 4, models (e.g., DUSt3R, CUT3R, Reloc3r, AC3D) fine-tuned on OmniWorld (including its pseudo-labeled subsets) achieve significant improvements on 3D Geometric Prediction and Video Generation benchmarks. These results empirically demonstrate that our annotations provide an effective and trustworthy supervisory signal.

---

> > ### Author Response · Authors · 2025-11-21
> >
> > **5. Provide New Analysis for Key Modalities.**
> > We conducted new analyses:
> >
> > **a) Depth.**
> > The accuracy of our OmniWorld-Game GT depth is validated by the experiments in Sec 4.1. For our pseudo-labeled DROID data (using FoundationStereo), we found that pre-training FP3 [4] model on point clouds projected from our refined depth yields significantly higher real-world task success rates compared to pre-training on the original DROID depth. The results are presented in terms of success rate in Table R1.
> >
> > **Table R1. Real-world manipulation task success rates.** Comparison between pre-training on original DROID depth vs. our refined depth.
> >
> > | Method | Open the drawer | Stack cups | Pick up the toy | Put the toy into basket |
> > | :--- | :---: | :---: | :---: | :---: |
> > | FP3 (original DROID depth) | $25$ | $10$ | $60$ | $35$ |
> > | **FP3 (our annotated DROID depth)** | $\mathbf{40}$ | $\mathbf{35}$ | $\mathbf{90}$ | $\mathbf{55}$ |
> >
> > **b) Camera Pose.**
> > For data without GT depth, our VGGT + CoTracker + BA pipeline outperforms the baseline VGGT on the Sintel benchmark significantly.
> >
> > **Table R2. Pose estimation performance on Sintel benchmark.** Comparison between the baseline VGGT and our full pipeline on data without GT depth.
> >
> > | Method | ATE $\downarrow$ | RPE trans $\downarrow$ | RPE rot $\downarrow$ |
> > | :--- | :---: | :---: | :---: |
> > | VGGT (Baseline) | $0.167$ | $0.062$ | $0.491$ |
> > | **Ours** | $\mathbf{0.082}$ | $\mathbf{0.042}$ | $\mathbf{0.246}$ |
> >
> > To evaluate the reliability of our camera pose annotations with GT depth, we followed the validation protocol proposed in [3]. We conducted a large-scale evaluation across $8,345$ randomly sampled pairs from our dataset.
> > For each pairs, we extracted high-quality sparse correspondences on static regions using SuperPoint and LightGlue, explicitly masking out dynamic objects. We then computed the geometric consistency (via Sampson error) for poses estimated by our pipeline compared to the DroidCalib baseline.
> >
> > As shown in **Table R3** below, our pipeline significantly outperforms the baseline. Notably, our method achieves a high level of accuracy, with 78.36\% of frame pairs exhibiting a reprojection error of less than 1 pixel (compared to 69.85\% for the baseline).
> >
> > **Table R3. Evaluation of camera pose annotation reliability.** We report geometric consistency (reprojection error) compared to the DroidCalib baseline.
> >
> > | Method | Mean Error (px) $\downarrow$ | % $< 1$Pix $\uparrow$ | % $< 3$Pix $\uparrow$ | % $< 5$Pix $\uparrow$ |
> > | :--- | :---: | :---: | :---: | :---: |
> > | DroidCalib (Baseline) | $1.30$ | $69.85\\%$ | $91.42\\%$ | $96.02\\%$ |
> > | **Ours** | $\mathbf{1.09}$ | $\mathbf{78.36\\%}$ | $\mathbf{93.90\\%}$ | $\mathbf{96.66\\%}$ |
> >
> > Beyond metrics, our pipeline produces visibly more consistent point cloud reconstructions than the baseline DroidCalib alone. Please refer to the newly added **Appendix F** for detailed visualizations.
> >
> > ---
> > [1] Jin et al., "Stereo4D: Learning How Things Move in 3D from Internet Stereo Videos", CVPR 2025.
> >
> > [2] Ling et al., "DL3DV-10K: A Large-Scale Scene Dataset for Deep Learning-Based 3D Vision", CVPR 2024.
> >
> > [3] Rockwell et al., "Dynamic Camera Poses and Where to Find Them", CVPR 2025.
> >
> > [4] Yang et al., "FP3: A 3D Foundation Policy for Robotic Manipulation", arXiv 2025.

---

> > > ### Comment · Reviewer_XxMY · 2025-11-23
> > > **Thank you for your detailed response**
> > >
> > > Thank you for your detailed response and for your hard work during the rebuttal. Your clarifications have resolved most of my concerns regarding the Sim-to-Real gap and overall dataset quality. I have just a few remaining questions:
> > >
> > > **1. Copyright and Licensing of OmniWorld-Game.**
> > > Since most OmniWorld-Game data is sourced from commercial video games, I still have concerns regarding copyright compliance. Although the paper includes an Ethics Statement, could the authors clarify whether you have obtained explicit permission from the respective game publishers to redistribute gameplay footage as part of a machine-learning dataset?
> > > If such permission has not been obtained, how do you ensure that releasing OmniWorld-Game will not violate copyright laws or the Terms of Service of those games?
> > >
> > > **2. Dataset Release Plan.**
> > > I noticed that many sub-datasets in the project repository are still not publicly available. In particular, OmniWorld-Game currently provides only around 5k sequences. Do the authors plan to fully release the complete dataset in the future? If so, could you clarify the expected timeline?

---

> > > > ### Author Response · Authors · 2025-11-24
> > > >
> > > > ### **Q1: Copyright and Licensing of OmniWorld-Game.**
> > > >
> > > > We sincerely thank the reviewer for the positive feedback on our rebuttal and for raising this critical question regarding copyright compliance. We take copyright seriously.
> > > >
> > > > **1. Adherence to Publisher Policies.** We have actively engaged with game publishers regarding the academic use and have reached explicit permission with a subset of them. Moreover, most modern game publishers publish "Terms of Service" that grant users permission to capture and share gameplay footage for non-commercial purposes. As stated in our Ethics Statement, we strictly align our practices with these policies. Our data collection utilizes open-source, non-invasive capture tools permitted by these policies, without any reverse engineering or hacking, which is standard practice for similar academic datasets like [1].
> > > >
> > > > **2. Strict Licensing and Access Control.** To further mitigate potential legal risks and respect IP rights, we release OmniWorld-Game under a strict CC BY-NC-SA 4.0 License with specific provisions. As detailed in our License, the dataset is released solely for academic research. Commercial use is expressly forbidden.
> > > >
> > > > ---
> > > > [1] Richter et al., "Playing for Data: Ground Truth from Computer Games", ECCV 2016.
> > > >
> > > > ### **Q2: Dataset Release Plan.**
> > > >
> > > > We thank the reviewer for their interest in the OmniWorld dataset. Regarding the release plan, we would like to provide the following details.
> > > >
> > > > **1. Phased Release Plan & Timeline.** We have a phased release plan to ensure the highest standards of data quality. We are currently conducting a final round of rigorous data cleaning and we aim to release all the data within the next three months.
> > > >
> > > > **2. Sufficiency and Effectiveness of the Current Release.** We wish to emphasize that the currently released subset is already a powerful resource on its own. The 5k sequences contain approximately 920K frames, which is already comparable to or even larger than many existing synthetic datasets (e.g., Sintel is ~1K frames, PointOdyssey is ~216K frames). To the best of our knowledge, our data has already been adopted by several influential works for either evaluating or training, demonstrating that even the current subset provides substantial value to the community while we prepare the full release.

---

> > > > > ### Comment · Reviewer_XxMY · 2025-11-25
> > > > > **Thank you for your response**
> > > > >
> > > > > Thank you for the detailed response. Your explanations regarding the technical aspects, data copyright, and release plan resolve my concerns. With these clarifications, I maintain my positive assessment of the paper.

---

> > > > > > ### Author Response · Authors · 2025-11-26
> > > > > >
> > > > > > We sincerely thank the reviewer for the positive feedback and for confirming that our response has resolved your concerns. We appreciate your time and valuable comments, which have greatly helped improve our work.

---

### Official Review · Reviewer_NEvh · 2025-11-01

**Soundness:** 2
**Presentation:** 3
**Contribution:** 2
**Rating:** 4
**Confidence:** 4

**Summary:**

This paper introduces a large-scale, multi-domain, multi-modal 4D dataset. It comprises over 214 hours of videos, includes simulation, human, robot and internet, and provides rich annotations including depth, camera, text, flow and masks. Based on this dataset, the paper also establishes a challenging 4D benchmark and validates the dataset as a powerful resource for training.

**Strengths:**

* The dataset is large-scale, multi-domain and multi-modal, while previous datasets are limited.
* The paper proposes an annotation pipeline for the data from different domains and different outputs.
* The paper validates that the dataset can be a powerful resource for both training and evaluation.

**Weaknesses:**

* It's hard to validate the quality of the dataset. The paper only leverages the pretrained models for annotation. The open-source real-world datasets are very noisy and the pretrained models can not handle them perfectly. How to filter the bad cases?
* The paper lacks the ablation analysis of the annotation pipeline. Which pretrained model is most suitable for each dataset and why.
* The paper lacks the analysis of how to select the benchmark samples.

**Questions:**

See Weakness.

---

> ### Author Response · Authors · 2025-11-21
>
> ### **Q1: Concerns regarding the quality of our pseudo-labels, the methods for handling and filtering bad cases.**
>
> We thank the reviewer for this critical question. We validate our dataset's quality through five key points: (1) a core clarification on our Ground Truth data, (2) a robust, multi-stage annotation pipeline, (3) efficacy validation via downstream task improvement, (4) multi-level filtering, and (5) new quantitative and qualitative analyses for key modalities.
>
> **1. We must clarify that our primary contribution, the OmniWorld-Game dataset, features pixel-perfect Ground Truth depth obtained directly from ReShade.**
> This is not a pseudo-label and fundamentally guarantees the accuracy of our core data. As shown in Table 1, OmniWorld-Game marks a significant leap in scale, featuring **18.5M frames with GT depth**, $18\times$ larger than classic synthetic datasets like TartanAir (1M frames), while uniquely offering dense annotations across depth, camera, text, flow, and Fg. masks. Its scale and high-precision ground truth make it an invaluable resource for the community. To the best of our knowledge, our data has already been adopted by several influential works for either evaluating or improving dynamic scene reconstruction, highlighting its effectiveness and growing influence.
>
> **2. A Customized, Multi-Stage Pipeline Ensures Quality.**
> We do not naively apply a single pre-trained model. Our pipeline (Sec 2.1, 2.2) is robustly designed:
> * **a) Strict Pre-filtering.** We apply "video slicing" to actively discard low-quality clips (e.g., motion blur, insufficient features) before annotation (Sec 2.1). For more detailed descriptions and bad case visualizations, please see the newly added **Appendix F**.
> * **b) Differentiated Annotation.** While OmniWorld-Game has GT depth, for other domains, we select SOTA models based on their specific data characteristics (e.g., stereo vs. sparse depth) and have rigorously validated their accuracy before deployment. Qualitative comparisons added to the **Appendix G** further confirm the superiority of our selected models over other methods.
> * **c) Multi-stage Optimization.** For dynamic camera pose, we employ a robust two-stage pipeline (e.g., DroidCalib/VGGT + CoTracker3 + BA). This optimization is proven to reduce errors [3].
>
> **3. Uncertainty is Handled via Multi-Level Filtering.**
> * **a) Source Filtering.** As mentioned, we filter low-quality clips pre-annotation (Sec 2.1, Appendix F).
> * **b) Post-filtering.** We will **open source** our data and dataloaders with built-in filtering rules (e.g., focal length range, depth thresholds, forward-backward optical flow consistency) to ensure users access clean data.
>
> **4. Downstream Performance Gains Validate Data Quality.**
> Following standard practice for large-scale datasets [1, 2], the most direct validation is the performance gain on downstream tasks. As shown in Sec. 4, models (e.g., DUSt3R, CUT3R, Reloc3r, AC3D) fine-tuned on OmniWorld (including its pseudo-labeled subsets) achieve significant improvements on 3D Geometric Prediction and Video Generation benchmarks. These results empirically demonstrate that our annotations provide an effective and trustworthy supervisory signal.

---

> > ### Author Response · Authors · 2025-11-21
> >
> > **5. Provide New Analysis for Key Modalities.**
> > We conducted new analyses:
> >
> > **a) Depth.**
> > The accuracy of our OmniWorld-Game GT depth is validated by the experiments in Sec 4.1. For our pseudo-labeled DROID data (using FoundationStereo), we found that pre-training FP3 [4] model on point clouds projected from our refined depth yields significantly higher real-world task success rates compared to pre-training on the original DROID depth. The results are presented in terms of success rate in Table R1.
> >
> > **Table R1. Real-world manipulation task success rates.** Comparison between pre-training on original DROID depth vs. our refined depth.
> >
> > | Method | Open the drawer | Stack cups | Pick up the toy | Put the toy into basket |
> > | :--- | :---: | :---: | :---: | :---: |
> > | FP3 (original DROID depth) | $25$ | $10$ | $60$ | $35$ |
> > | **FP3 (our annotated DROID depth)** | $\mathbf{40}$ | $\mathbf{35}$ | $\mathbf{90}$ | $\mathbf{55}$ |
> >
> > **b) Camera Pose.**
> > For data without GT depth, our VGGT + CoTracker + BA pipeline outperforms the baseline VGGT on the Sintel benchmark significantly.
> >
> > **Table R2. Pose estimation performance on Sintel benchmark.** Comparison between the baseline VGGT and our full pipeline on data without GT depth.
> >
> > | Method | ATE $\downarrow$ | RPE trans $\downarrow$ | RPE rot $\downarrow$ |
> > | :--- | :---: | :---: | :---: |
> > | VGGT (Baseline) | $0.167$ | $0.062$ | $0.491$ |
> > | **Ours** | $\mathbf{0.082}$ | $\mathbf{0.042}$ | $\mathbf{0.246}$ |
> >
> > To evaluate the reliability of our camera pose annotations with GT depth, we followed the validation protocol proposed in [3]. We conducted a large-scale evaluation across $8,345$ randomly sampled pairs from our dataset.
> > For each pairs, we extracted high-quality sparse correspondences on static regions using SuperPoint and LightGlue, explicitly masking out dynamic objects. We then computed the geometric consistency (via Sampson error) for poses estimated by our pipeline compared to the DroidCalib baseline.
> >
> > As shown in **Table R3** below, our pipeline significantly outperforms the baseline. Notably, our method achieves a high level of accuracy, with 78.36\% of frame pairs exhibiting a reprojection error of less than 1 pixel (compared to 69.85\% for the baseline).
> >
> > **Table R3. Evaluation of camera pose annotation reliability.** We report geometric consistency (reprojection error) compared to the DroidCalib baseline.
> >
> > | Method | Mean Error (px) $\downarrow$ | % $< 1$Pix $\uparrow$ | % $< 3$Pix $\uparrow$ | % $< 5$Pix $\uparrow$ |
> > | :--- | :---: | :---: | :---: | :---: |
> > | DroidCalib (Baseline) | $1.30$ | $69.85\\%$ | $91.42\\%$ | $96.02\\%$ |
> > | **Ours** | $\mathbf{1.09}$ | $\mathbf{78.36\\%}$ | $\mathbf{93.90\\%}$ | $\mathbf{96.66\\%}$ |
> >
> > Beyond metrics, our pipeline produces visibly more consistent point cloud reconstructions than the baseline DroidCalib alone. Please refer to the newly added **Appendix F** for detailed visualizations.
> >
> > ---
> > [1] Jin et al., "Stereo4D: Learning How Things Move in 3D from Internet Stereo Videos", CVPR 2025.
> >
> > [2] Ling et al., "DL3DV-10K: A Large-Scale Scene Dataset for Deep Learning-Based 3D Vision", CVPR 2024.
> >
> > [3] Rockwell et al., "Dynamic Camera Poses and Where to Find Them", CVPR 2025.
> >
> > [4] Yang et al., "FP3: A 3D Foundation Policy for Robotic Manipulation", arXiv 2025.

---

> ### Author Response · Authors · 2025-11-21
>
> ### **Q2: Lack of ablation analysis and rationale for model selection.**
>
> We thank the reviewer for the constructive comment regarding the rationale for model selection. In our design, we prioritized robust, modality-specific performance.
>
> **1. Modality-Specific Optimization & Multi-Stage Validation.**
> We adopted a practical strategy of selecting off-the-shelf SOTA models tailored to the specific modality of each dataset, whose superior performance has already been rigorously established in their respective publications. To ensure their suitability for OmniWorld, we conducted preliminary comparisons to identify the most robust option. The effectiveness of these choices is ultimately confirmed by the functional validation in **Sec. 4**. Qualitative comparisons added to the newly added **Appendix G** further confirm the superiority of our selected models over other methods.
>
> **2. Detailed Justification by Modality.**
> * **a) Depth.** We matched models to data types: FoundationStereo for the stereo dataset DROID (stereo depth SOTA, validated by our FP3 experiments), and PriorDA (prior depth SOTA) for datasets with sparse priors, aligning with high-quality datasets like ViPE [2].
> * **b) Camera Pose.** We employed a robust Coarse-to-Fine pipeline (VGGT/DroidCalib initialized, refined via CoTracker3 + Bundle Adjustment). This strategy balances efficiency with geometric precision, which has been proved in [1, 3].
> * **c) Foreground Masks.** We adopted the standard Grounding DINO + SAM pipeline [2, 3]. Our internal comparisons (added to Appendix G) demonstrate that this approach yields superior segmentation over video-specific models like SegAnyMo [4] in our complex dynamic scenarios.
> * **d) Optical Flow.** DPFlow was selected specifically for its capability to process videos at original high resolutions without downsampling.
>
> Each model was evaluated on a randomly selected unlabeled dataset, after which human raters assessed the results to select the best-performing models.
>
> ---
> [1] Aether Team et al., "Aether: Geometric-Aware Unified World Modeling", ICCV 2025.
>
> [2] Huang et al., "VIPE: Video Pose Engine for 3D Geometric Perception", arXiv 2025.
>
> [3] Rockwell et al., "Dynamic Camera Poses and Where to Find Them", CVPR 2025.
>
> [4] Huang et al., "Segment Any Motion in Videos", CVPR 2025.

---

> ### Author Response · Authors · 2025-11-21
>
> ### **Q3: Lack of analysis on benchmark sample selection.**
>
> We appreciate the reviewer raising this point. We agree that a transparent sample selection strategy is fundamental to ensuring the benchmark's representativeness.
> To address the limitations of existing benchmarks (e.g., short sequences, static scenes), we employed an "Attribute-Balanced Sampling" strategy to ensure comprehensive coverage of real-world complexities.
>
> **1. Sampling for 3D Geometric Prediction:**
> We curated long-sequence samples to rigorously test generalization across diverse environments and dynamics:
> * **Scene Distribution:** Balanced across Outdoor-Urban ($\sim 58\\%$, e.g., streets, plazas), Outdoor-Natural ($\sim 18\\%$, e.g., forests, cliffs), and Indoor ($\sim 24\\%$) to test both bounded and unbounded scenarios.
> * **Dynamic Complexity:** Heavily weighted towards High-Dynamic ($\sim 48\\%$, e.g., running, flying) and Medium-Dynamic ($\sim 38\\%$, e.g., walking, standard interactions) content to address the lack of high-amplitude motion in datasets like Sintel.
> * **Sequence Length:** All samples are extended $384$-frame ($16\text{s}$) sequences, specifically selected to evaluate long-term geometric consistency.
>
> **2. Sampling for Camera-Controlled Video Generation:**
> We prioritized trajectory complexity and environmental richness:
> * **Camera Trajectory Complexity:** Split between High-Dynamic ($\sim 51\\%$, featuring rapid translation, rotation, or compound high-speed movements) and Medium-Dynamic ($\sim 49\\%$, distinct but stable motion) to challenge control adherence.
> * **Diversity:** Spans varied lighting (Day $60\\%$, Dusk $9\\%$, Night $31\\%$), weather (Sunny $53\\%$, Cloudy $25\\%$, Rain/Snow $22\\%$), and perspectives (1st-person $64\\%$, 3rd-person $36\\%$).
>
> We have added a new **Appendix I** to provide a comprehensive description and visualization of our benchmark details.

---

> > ### Comment · Reviewer_NEvh · 2025-11-28
> >
> > Thanks for the detailed response. The pipeline of the dataset construction is very comprehensive. I would raise my point.

---

> > > ### Author Response · Authors · 2025-11-28
> > >
> > > Thank you for your positive feedback and for raising the score! We are glad that our response addressed your concerns. We appreciate your time and valuable comments, which have greatly helped improve our work.

---

### Author Response · Authors · 2025-11-21

We are grateful to Reviewers NEvh, XxMY, and xQoM for their time and helpful feedback. Your comments have played a key role in improving our paper.

Regarding the reviewers' common concerns about data quality validation, annotation pipeline rationale, and Sim-to-Real Gap, we have conducted extensive new analyses. We have verified the effectiveness of our annotation pipeline through additional experiments and demonstrated that the Sim-to-Real gap can be effectively bridged with our OmniWorld-Game.

We have uploaded a revised draft of our paper with new **Appendices (F-I)**, covering video clip filtering, qualitative annotation validation, failure case analysis, and detailed benchmark statistics.

Your reviews helped us make the work more complete. We have posted individual responses to each of your comments below.

---

> ### Comment · Reviewer_xQoM · 2025-11-27
>
> Thank you for your thoughtful replies and for carefully reviewing each reviewer's questions.

---

### Author Response · Authors · 2025-12-01
**Summary**

Dear Reviewers and AC,

We sincerely appreciate the time and effort you have dedicated to reviewing our manuscript. During the rebuttal phase, all three reviewers (NEvh, XxMY, xQoM) expressed satisfaction with our responses and manuscript revisions. We summarize the rebuttal as follows.

**1. Consensus on Key Issues.**
* **Dataset Quality & Pipeline (Reviewer NEvh, XxMY):** We addressed concerns regarding annotation quality by providing additional qualitative and quantitative validation and clarifying our robust pipeline. Both reviewers agreed that these concerns are resolved. Reviewer NEvh specifically acknowledged this, stating the pipeline is "very comprehensive" and indicated an intention to "raise my point."
* **Sim-to-Real & Generalization (Reviewer XxMY, xQoM):** We demonstrated the effectiveness of OmniWorld-Game in bridging the Sim-to-Real gap and clarified our systematic experimental analysis of multi-domain data. Both reviewers confirmed their concerns were resolved, with Reviewer xQoM recognizing the work as "solid."

**2. Summary of Manuscript Revisions.**
Based on the feedback, we have carefully revised and enhanced the manuscript as follows (changes marked in blue):
* New Appendix F: Details on high-quality video clip filtering.
* New Appendix G: Qualitative validation of our annotation pipelines.
* New Appendix H: Failure case analysis.
* New Appendix I: Detailed benchmark statistics.
* New Appendix J: Quantitative validation of our annotation pipelines.

We hope these updates help better convey the value of the OmniWorld dataset to the community.

Best Regards,

Authors.

---

### Meta-Review · Area_Chair_TvqM · 2026-01-07

**Summary:**

This paper receives 1x marginal reject and 2x marginal accepts at the initial review. The strengths of the paper are: 1) dataset is large-scale, multi-domain and multi-modal, which is a powerful resource for training and validation. 2)  Cross-domain integration and automated annotation pipeline. 3) Demonstrated training value. 4) Challenging benchmarks are constructed for 3D geometry prediction and camera control video generation. The weaknesses are: 1) hard to validate the quality of the dataset. 2)  lacks the ablation analysis of the annotation pipeline 3)  lacks the analysis of how to select the benchmark samples. 4) Unclear handling of dynamic 4D content 5) High reliance on synthetic data and limited real-world evaluation. 6) Copyright and Licensing of OmniWorld-Game. The issues are mostly clarified during the rebuttal and discussion phase. The major concern of IP issue raised by the second reviewer is also resolved during the discussion phase. The reviewer who gave the marginal reject is satisfied with the clarification and mentioned that the score would be raised. The other two reviewers are also satisfied with the rebuttal and discussion and seem to maintain their positive reviews. The AC follows the suggestions of the reviewers to accept the paper.

**Reviewer Concerns:**

The weaknesses are: 1) hard to validate the quality of the dataset. 2)  lacks the ablation analysis of the annotation pipeline 3)  lacks the analysis of how to select the benchmark samples. 4) Unclear handling of dynamic 4D content 5) High reliance on synthetic data and limited real-world evaluation. 6) Copyright and Licensing of OmniWorld-Game. The issues are mostly clarified during the rebuttal and discussion phase. The major concern of IP issue raised by the second reviewer is also resolved during the discussion phase. The reviewer who gave the marginal reject is satisfied with the clarification and mentioned that the score would be raised. The other two reviewers are also satisfied with the rebuttal and discussion and seem to maintain their positive reviews.

**Reviewer Scores:**

The reviewer who gave the marginal reject is satisfied with the clarification and mentioned that the score would be raised. The other two reviewers are also satisfied with the rebuttal and discussion and seem to maintain their positive reviews.

---

### Decision · Program_Chairs · 2026-01-26

Accept (Poster)